# Actomyosin polarisation through PLC-PKC triggers symmetry breaking of the mouse embryo

Meng Zhu[1], Chuen Yan Leung[1], Marta N. Shahbazi[1] & Magdalena Zernicka-Goetz[1]

Establishment of cell polarity in the mammalian embryo is fundamental for the first cell fate decision that sets aside progenitor cells for both the new organism and the placenta. Yet the sequence of events and molecular mechanism that trigger this process remain unknown. Here, we show that de novo polarisation of the mouse embryo occurs in two distinct phases at the 8-cell stage. In the first phase, an apical actomyosin network is formed. This is a pre-requisite for the second phase, in which the Par complex localises to the apical domain, excluding actomyosin and forming a mature apical cap. Using a variety of approaches, we also show that phospholipase C-mediated $PIP_2$ hydrolysis is necessary and sufficient to trigger the polarisation of actomyosin through the Rho-mediated recruitment of myosin II to the apical cortex. Together, these results reveal the molecular framework that triggers de novo polarisation of the mouse embryo.

[1] Mammalian Embryo and Stem Cell Group, Department of Physiology, Development and Neuroscience, University of Cambridge, Downing Street, Cambridge CB2 3EG, UK. Correspondence and requests for materials should be addressed to M.Z.-G. (email: mz205@cam.ac.uk)

Cell polarisation leading to the asymmetric distribution of cellular components is critical for cell fate specification and cellular rearrangements during development, as well as for the maintenance of adult tissue homeostasis[1–4]. In contrast to the development of embryos of many species, mammalian embryos acquire cell polarity de novo at a species-specific developmental stage. In the mouse embryo, cell polarisation becomes established between the second and third day after fertilisation, at the 8-cell stage, resulting in defined apical and basolateral domains[5, 6]. Consistent with canonical apicobasal polarisation, the apical domain becomes enriched with the Par3-Par6-aPKC complex, while the basolateral domain becomes enriched with cell adhesion proteins[7–9]. This acquisition of cell polarity coincides with embryo compaction, which leads to a tighter embryonic geometry as a consequence of cell–cell contact elongation and sealing of adjacent blastomeres[10, 11].

Establishment of cell polarity at the 8-cell stage is a critical morphogenetic event, as the presence of the apical polarity domain directs the first bifurcation of extra-embryonic and embryonic lineages during the next cell divisions[12]. The cells that inherit the apical domain are specified as trophectoderm (TE), which will give rise to the placenta, while the cells that lack the apical domain maintain pluripotency and develop as inner cell mass, which will give rise to the foetus and yolk sac[13]. Consequently, defective polarisation leads to altered cell fate specification, failure of blastocyst formation and developmental arrest[14, 15]. Despite its major importance, it remains unknown how cell polarity becomes first established in the mammalian embryo.

Here, we demonstrate that cell polarisation in the mouse embryo is initiated by PLC-mediated $PIP_2$ hydrolysis that activates protein kinase C (PKC), and in turn RhoA, leading to cortical accumulation of actomyosin. By using a variety of approaches to eliminate PKC function and optogenetic techniques to activate it locally, we show that ectopic activation of PKC is sufficient to give a local enrichment of actin and phosphorylated myosin light chain. Induction of this cytoskeletal asymmetry is an absolute pre-requisite for the cortical enrichment of the Par complex to establish cell polarity and form a mature apical cap. These findings provide a molecular framework for how the reorganisation of the actomyosin network triggers cell polarisation in a temporally controlled manner in the mouse embryo.

## Results

### Actomyosin and Par complex dynamics define two phases of cell polarisation during mouse embryogenesis.

The actomyosin network and the Par complex represent two conserved systems used to establish cell polarity in many model systems[3, 16–18]. We therefore first wished to determine the behaviour of these molecular complexes as cell polarity becomes established and as cells compact in the mouse embryo. To this end, we examined their localisation from the early to the late 8-cell stage using the angle between adjacent blastomeres (inter-blastomere angle or IEA) as a measure of the extent of compaction, and thus temporal progression through the 8-cell stage, and F-actin and Pard6 as respective markers of actomyosin and the Par complex (Supplementary Fig. 1a). We found that actomyosin and the Par complex became polarised following a step-wise pattern. Analysis of the circumferential distribution of F-actin and Pard6 in cell-contact and cell-contact-free regions revealed that during the early 8-cell stage (within 1 h post cell division) to mid 8-cell stage (3–4 h post cell division), F-actin became gradually localised apically, while Pard6 remained distributed equally around the cell cortex (Fig. 1a–c and Supplementary Fig. 1b). Only at the mid-late 8-cell stage (5–8 h post cell division), did Pard6 begin to accumulate apically. Upon apical enrichment of Pard6, F-actin

became redistributed to surround the Par apical domain in a ring-like structure (Fig. 1a–c). To confirm this sequence of events, we also examined the localisation of PKCζ, another essential component of the Par complex (Supplementary Fig. 1c).

As an alternative readout of actomyosin localisation, we injected messenger RNA (mRNA) for GFP-tagged myosin regulatory light chain (GFP-MRLC) at the 2-cell stage and followed subsequent development. In agreement with the above observation, this revealed that GFP-MRLC became enriched at the apical domain much earlier than Pard6, and subsequently redistributed to the periphery of the Pard6 domain upon Pard6 apical enrichment (Supplementary Fig. 1d–f). To further confirm that the apical enrichment of actomyosin precedes apical domain maturation, we compared the localisation of F-actin during the 8-cell stage to the localisation of another two conserved apical domain markers, Crb3[19] and Ezrin[20] by double-immunostaining or time-lapse imaging. In both cases, we found that F-actin polarised to the cell-contact-free surface before any apical markers (Supplementary Fig. 1g–i). Collectively, these results indicate that de novo polarisation of the mouse embryo occurs in two phases: in the first phase, the actomyosin network polarises to the cell-contact-free surface during embryo compaction. In the second phase, Par proteins, together with other conserved apical components, become polarised and the actomyosin network forms a ring structure around the Par-enriched domain once compaction is completed (Fig. 1d).

### Asymmetry of the functional actomyosin network is required for Par complex polarisation.

Since the above observations indicated that apical localisation of the actomyosin network precedes localisation of Par complex components, we next wished to determine the functional requirement of actomyosin asymmetry for the apical enrichment of the Par complex. To this end, we first attempted to use RNAi to deplete MRLC as a way of inhibiting actomyosin activity. We found that although we were able to deplete over 74% of the mRNA of the predominant MRLC isoform expressed at the early cleavage stages[21, 22], its protein levels remained largely unchanged, most likely reflecting a large maternal storage (Supplementary Fig. 2a, b). To circumvent this difficulty, we turned to use pharmacological inhibitors. As contractility has been viewed as a general function of actomyosin in the regulation of cellular morphogenesis[23] and recently it has been suggested to be required for compaction[24], we first sought to investigate whether actomyosin contractility is required for constructing the apical domain. To this end, we used blebbistatin, a potent and selective inhibitor of myosin II ATPase activity[25]. We applied blebbistatin at the beginning of the 8-cell stage, to avoid inhibiting the preceding cytokinesis, and determined cell polarity by examining the localisation of Pard6 and F-actin, and compaction by measuring IEA. We found that although blebbistatin treatment abolished compaction, establishment of apical domain remained unaffected, as both F-actin and Pard6 remained polarised to the cell-contact-free domain (Fig. 2a–d). The result was the same regardless of whether we increased the blebbistatin concentration or extended the timing of the treatment (Supplementary Fig. 2c), suggesting that actomyosin contractility, although critical for compaction, is dispensable for establishment of the apical domain.

To determine whether apical enrichment of actomyosin is required for Par complex, we treated 4–8-cell stage embryos with ML-7 (an inhibitor of MLCK[26]) or Y-27632 (an inhibitor of ROCK[27]) to abolish the phosphorylation-mediated assembly of myosin II filament, or latrunculin B (LatB) to potently destroy the actin meshwork[28]. We found that, contrary to blebbistatin treatment, all three inhibitors significantly abolished Pard6's

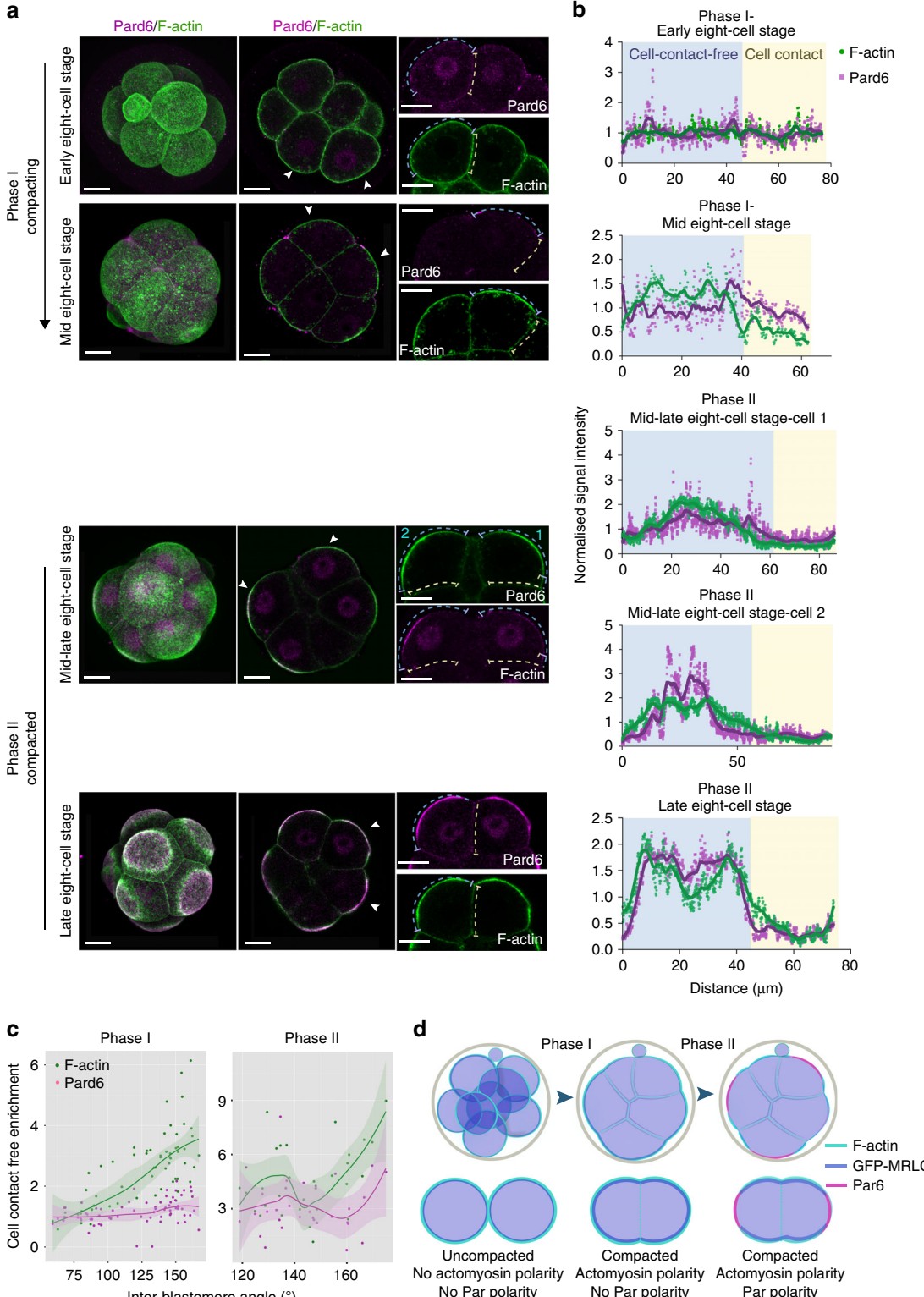

**Fig. 1** Dynamics of actomyosin and Pard6 polarisation at the 8-cell stage. **a** Mouse embryos were fixed at early, mid and late 8-cell stage, and immunostained for F-actin and Pard6. Arrowheads indicate the magnified blastomeres. **b** Circumferential line profiles of F-actin and Pard6 fluorescence intensity in embryos from (**a**). Yellow indicates cell–cell contact regions and blue cell-contact-free domain. Smoothened data were displayed as coloured curves to show the pattern of the signal. **c** Cell-contact-free enrichment of F-actin and Pard6 plotted against the IEA in embryos from (**a**). Each dot represents an individual measurement. $N = 25$ embryos, three independent experiments. LOWESS plot were used to display the tendency of different datasets. The shadow indicates the 95% confidence interval. **d** Summary of the different phases of polarisation during the 8-cell stage. During the first phase, embryos start compacting and actomyosin becomes polarised to the apical domain. During the second phase, embryos are already compacted and apical domain components, including Pard6 polarise apically. All scale bars, 15 μm

cell-contact-free enrichment (Fig. 2a–d and Supplementary Fig. 2d). ML-7 and LatB treatment also inhibited compaction and actomyosin asymmetry (Fig. 2a–d). To confirm the effects of the inhibitors, we examined the pattern of phosphorylated MRLC or F-actin after different inhibitor treatments. We found that, at the late 8-cell stage, a similar level of phosphorylated MRLC remained enriched at the cell-contact-free domain in blebbistatin-treated compared to control blastomeres, indicating that blebbistatin did not affect actomyosin activation or polarisation (Supplementary Fig. 2e, f). On the contrary, both ML-7 and

Y-27632 treatment led to a reduction of phosphorylated MRLC at the cell-contact-free surface, with a stronger decrease resulting from ML-7 treatment (Fig. 2e, f and Supplementary Fig. 2g, h). LatB treatment strongly disturbed F-actin organisation and resulted in its patchy distribution (Fig. 2a).

To unequivocally determine whether the apically polarised actomyosin is indeed essential for Par complex recruitment as the above results suggested, we next wished to destabilise actomyosin structure after actomyosin had become polarised and before the recruitment of Par complex. To this end, we applied ML-7 at mid

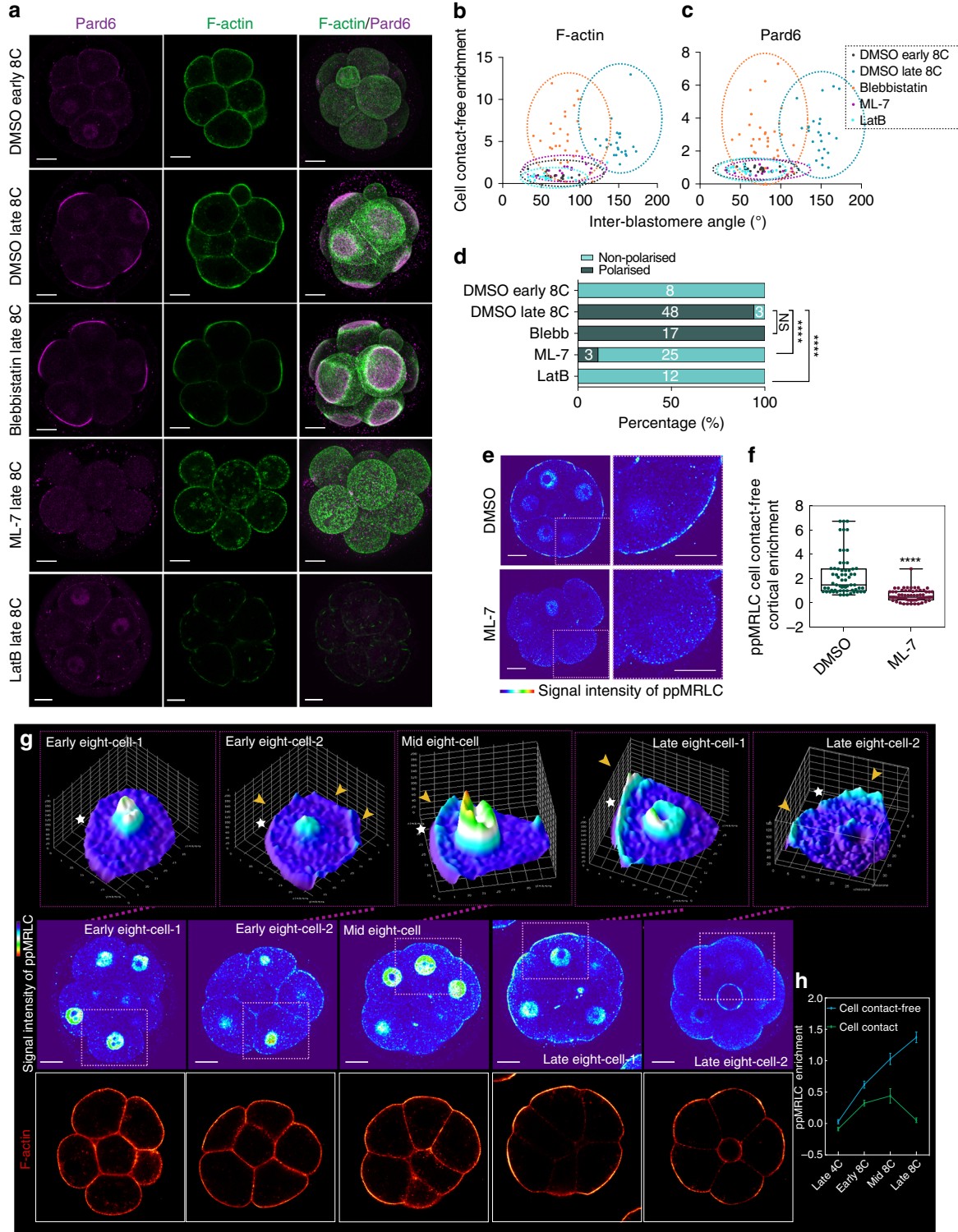

8-cell stage (3–4 h post cell division when embryos were compacted) and determined the localisation of actomyosin and the Par complex at the late 8-cell stage. We also measured the IEA to monitor the change of compaction status. We found that delayed ML-7 treatment did not affect compaction, as the IEA remained similar between the ML-7- and DMSO (control)-treated groups (Supplementary Fig. 2i, j). However, the enrichment of GFP-MRLC at the cell-contact-free domain was abolished in the ML-7-treated group (Supplementary Fig. 2i, k), indicating the effectiveness of delayed ML-7 treatment in disturbing actomyosin organisation. More importantly, the recruitment of Pard6 to the cell-contact-free surface was strongly inhibited (Supplementary Fig. 2i, l). Taken together, these results indicate that the intact organisation of an apically polarised actomyosin network, which pioneers apical domain formation, is required for Par polarisation.

**Activation of myosin triggers cell polarisation at the early 8-cell stage.** How cell polarity is specifically induced at the 8-cell stage remains unknown. Since our results indicated that the polarised organisation of actomyosin precedes and is required for polarisation of the Par complex, we next wished to determine how and when the actomyosin network becomes polarised. Given that myosin activation is required for actomyosin polarisation (Fig. 2a–d), we considered that myosin could be activated and polarised specifically at the 8-cell stage or, alternatively, myosin could be active throughout pre-implantation development, but at the 8-cell stage, an asymmetric inhibitory signal could lead to its polarised activity. To address this question, we examined the localisation of phosphorylated MRLC in relation to F-actin from the 4-cell to the late 8-cell stage. We found that the first cortical localisation of phosphorylated MRLC could be detected at the early 8-cell stage, but not at any time throughout 4-cell stage (Fig. 2g, h and Supplementary Fig. 2m, n). By the mid 8-cell stage, phosphorylated MRLC became enriched at the cell-contact-free cortex of all 8-cell blastomeres in a similar fashion to GFP-MRLC (Fig. 2g, h and Supplementary Fig. 1d–f). The level of phosphorylated MRLC progressively increased until the late 8-cell stage, at which point phosphorylated MRLC formed a ring-like structure and co-localised with the cortical meshwork of F-actin (Fig. 2g, h). These results indicate that the activation and polarisation of myosin network initiates cell polarisation at the early mid 8-cell stage.

**PLC-mediated PKC activity is critical for cell polarisation.** PKC family members are known to play a role in compaction[29, 30]. However, whether PKC is also required to establish cell polarity in the mouse embryo remains currently unknown. To test if PKC is required for cell polarisation, we wished to inhibit all the

multiple PKC isoforms expressed in mouse embryos at this stage[31]. To this end, we used two pan-PKC inhibitors highly selective for conventional PKCs (cPKCs) and novel PKCs[32, 33], sphingosine and calphostin C, and applied them to embryos at the 4- to 8-cell stage transition. We found that both sphingosine and calphostin C treatments prevented compaction as well as the cell-contact-free surface enrichment of F-actin and Pard6 (Fig. 3a–f), suggesting that PKC is important not only for compaction but also for cell polarisation in the mouse embryo.

The canonical activation of cPKCs requires both diacylglycerol (DAG) and inositol triphosphate (IP3), which in turn triggers intracellular calcium release required for cPKC activation[34]. DAG and IP3 are the products of the hydrolysis of the membrane phospholipid PtdIns(4,5)P2 (PIP$_2$), a process catalysed by PLC. To test if this pathway is responsible for PKC activation, we prevented DAG production from PIP$_2$ hydrolysis by applying a highly selective pan-PLC inhibitor (U73122)[35] from the 4-cell stage onwards, and assessed cell polarity at the late 8-cell stage (Fig. 3g). In agreement with findings above, U73122 treatment prevented the establishment of the apical domain and resulted in compaction failure (Fig. 3h–l; Supplementary Fig. 3a–c and Supplementary Movie 1, 2). Although establishment of cell polarity was inhibited, blastomeres were still able to undergo cytokinesis (Supplementary Fig. 3a, d, e). At the 16-cell stage, polarisation of the outer cells is involved in activating expression of the key trophectoderm transcription factor, Cdx2[12, 36]. To examine whether cell polarisation was functionally hindered and so affected cell fate specification and differentiation into the trophectoderm lineage, we examined the expression of Cdx2. We found that U73122-treatment abolished Pard6 polarisation and Cdx2 expression, in contrast to control embryos in which outer cells showed polarised Pard6 and Cdx2 expression as previously described[37, 38] (Supplementary Fig. 3f, g).

To determine whether the effect of PLC inhibition is the specific result of DAG depletion and lack of PKC activation, we co-applied the PLC inhibitor U73122 and the DAG analogue and PKC activator, 1-oleoyl-2-acetyl-sn-glycerol (OAG)[39] to determine if this would rescue the observed phenotype. We found that in all embryos co-treated with OAG, blastomeres became polarised, indicating that the defect in apical domain formation caused by PLC inhibition was rescued by the activation of PKC (Fig. 3h–l).

To further confirm that PLC-catalysed PIP$_2$ hydrolysis acts upstream of PKC to trigger cell polarisation, we used a dominant-negative genetic construct as an alternative way to directly inhibit PIP$_2$ hydrolysis. We overexpressed the fluorescent-tagged PH domain of PLCδ1 (Plcd1-PH-Ruby), a dominant-negative form of PLC isotypes (PLC-DN), which upon overexpression blocks PIP$_2$ hydrolysis by PLC as a result of competitive binding to

**Fig. 2** A polarised and active actomyosin network is required for Par complex polarisation. **a** 8-cell stage (blebbistatin, LatB group and its control) or 4–8 cell stage embryos (ML-7 and its control) were treated with DMSO, blebbistatin, ML-7 or LatB, and fixed at the late 8-cell stage. Early 8-cell stage embryos treated with DMSO were also fixed as a control. Embryos were immunostained for F-actin and Pard6. **b, c** Cell-contact-free enrichment of F-actin (**b**) or Pard6 (**c**) as a function of the IEA in embryos from (**a**). Each dot represents an individual measurement. **d** Percentage of polarised embryos in the groups of (**a**). The number of embryos per category is indicated. Embryos that did not show any blastomere with the enrichment of F-actin and Pard6 at the cell-contact-free surface were considered non-polarised. Otherwise embryos were classified as polarised. Data are shown as a contingency table. ****$p < 0.0001$, NS = not significantly different, Fisher's exact test (six independent experiments). **e** DMSO and ML-7-treated embryos immunostained for ppMRLC. **f** Quantification of ppMRLC cell-contact-free cortical enrichment in ML-7-treated and control embryos. Each dot represents an individual measurement. Data are shown as individual data points with Box and Whiskers graph (bottom: 25%; top: 75%; line: median; whiskers: min to max). ****$p < 0.0001$, Mann–Whitney test. $N = 10$ embryos for DMSO and $N = 12$ embryos for ML-7, four independent experiments. **g** ppMRLC and F-actin staining from the late 4-cell stage to the late 8-cell stage. The ppMRLC signal intensity of the indicated blastomeres is plotted in 3D. White stars indicate the cell-contact-free surface. Arrowheads indicate cortical localisation of ppMRLC. **h** Quantification of ppMRLC enrichment at the cell-contact and cell-contact-free domains from the late 4-cell stage to the late 8-cell stage. Data are shown as mean ± s.e.m. For all panels, squares indicate the magnified regions. $N = 32$ measurements from 7 late 4-cell stage embryos; $N = 69$ measurements from 11 early 8-cell stage embryos; $N = 86$ measurements from 11 mid 8-cell stage embryos; $N = 50$ measurements from 13 late 8-cell stage. All scale bars, 15 μm

PIP$_2$[40, 41]. PLC-DN mRNA was injected into both blastomeres at the 2-cell stage (and only Ruby mRNA in control embryos) and the localisation of F-actin and Pard6 were examined at the late 8-cell stage (Fig. 3m). We found that overexpression of PLC-DN inhibited formation of the apical domain at the late 8-cell stage (Fig. 3n–p). Despite the polarisation failure, cytokinesis still took place (Supplementary Fig. 4a, b). Overexpression of PLC-DN also

blocked compaction (Fig. 3n; Supplementary Fig. 4a and Supplementary Movie 3, 4), consistent with the pharmacological inhibition of PLC (Fig. 3i, j and Supplementary Fig. 3a). An equivalent impairment of Cdx2 expression was observed in PLC-DN overexpressing embryos at the 16-cell stage, indicating that apical polarisation is functionally abolished by PLC-DN (Supplementary Fig. 4c, d). To determine the specificity of the PLC-DN,

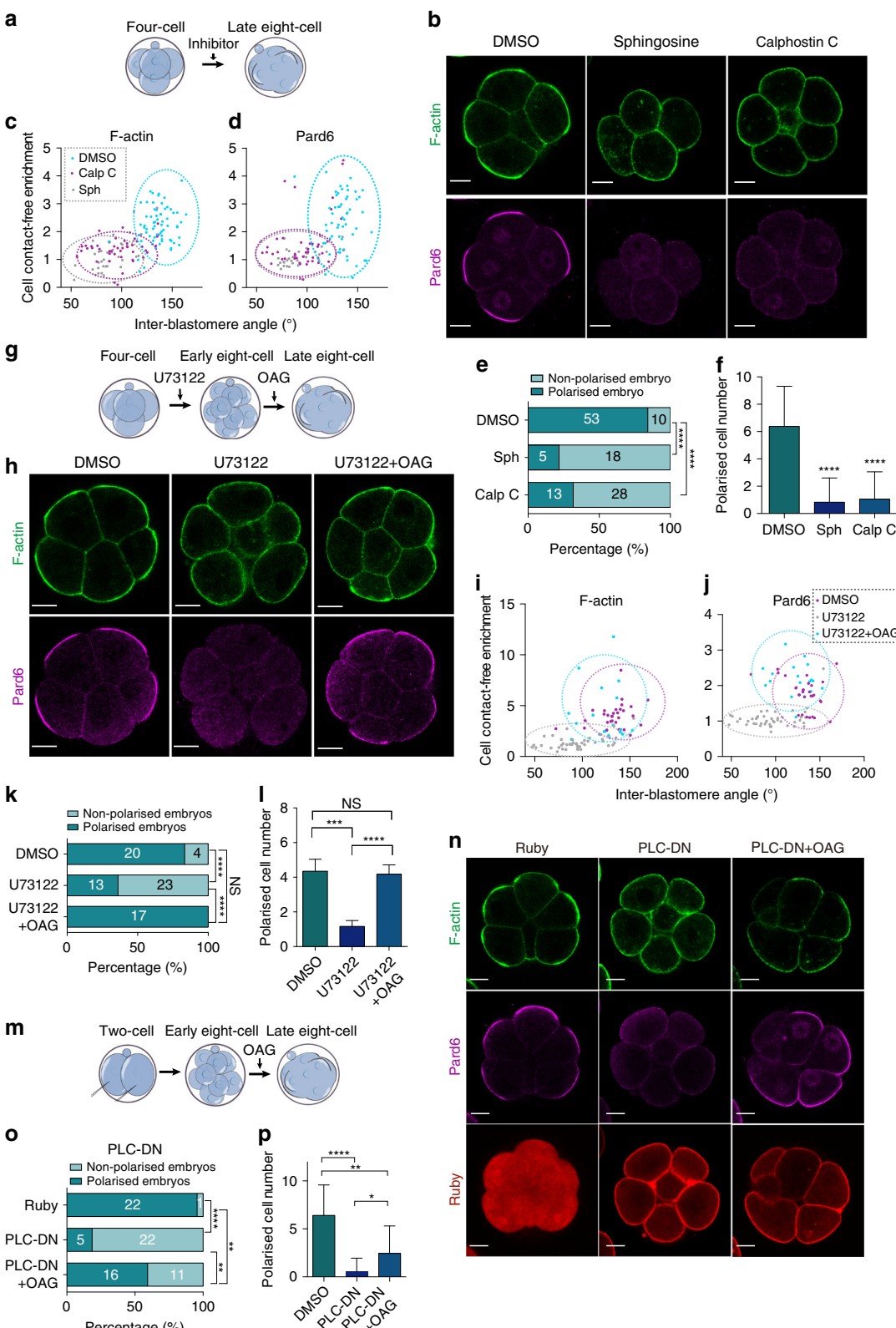

we also carried out a rescue experiment by applying the PKC activator OAG at the early 8-cell stage to PLC-DN expressing embryos. We found that OAG treatment was able to partially rescue the polarity defect arising from PLC-DN expression (Fig. 3n–p).

The above results indicated that the PLC–PKC signalling pathway and MRLC phosphorylation are required for establishing cell polarity at the 8-cell stage. However, whether MLCK-mediated phosphorylation of MRLC is triggered downstream of the PLC–PKC activation remained unknown. To address this question, we next examined the distribution of phosphorylated MRLC in PLC-DN expressing embryos and also in 8-cell embryos treated with sphingosine, calphostin C or U73122. We found that the level of phosphorylated MRLC at the cell-contact-free surface decreased following all of these treatments when compared to control embryos (Fig. 4a–h). Together, these results indicate that PLC–PKC signalling controls the level of phosphorylated MRLC on the apical domain in the first phase of cell polarisation.

**Ectopic PKC activation induces premature actomyosin polarisation but not polarisation of Par complex components**. Since our results demonstrated that PLC–PKC signalling is required for the establishment of cell polarity at the 8-cell stage, we next wished to determine whether upregulating PKC activity would be sufficient to augment cell polarisation. To this end, we used two different methods to activate PKC: treatment with the PKC activator OAG and overexpression of constitutively activate PKCα (PKCα-A25E)[42]. We applied OAG at the early 8-cell stage and found that while in control embryos Pard6 was restricted to the central part of the contact-free surface (mean surface coverage ratio (SCR) of 57.3%; Fig. 5a, b), treatment with OAG led to a significant increase in Pard6 SCR (84.8%) at the late 8-cell stage, and in certain cases, Pard6 even expanded throughout the entire contact-free surface (Fig. 5a, b). Prolonged OAG treatment up to the 16-cell stage induced an expansion of the Pard6 domain to the whole cell membrane, not only of outer but also of inner cells (Supplementary Fig. 5a, b). To determine if PKC activation could induce premature cell polarisation before 8-cell stage, we applied OAG at the early 4-cell stage. OAG treatment led to premature compaction, as previously described[42], and resulted in the enrichment of both F-actin and phosphorylated MRLC at the cell-contact-free surface (Fig. 5c, e). However, although actomyosin was robustly polarised, no enrichment of Pard6 at the cell-contact-free surface could be observed (Fig. 5c–e and Supplementary Fig. 5c). A similar result was observed following PKCα-A25E overexpression, which induced a polarised distribution of F-actin and GFP-MRLC towards the cell contact-free surface but not of Pard6 (Fig. 5f–h and Supplementary Fig. 5d). This indicates that although PKC activity is sufficient to initiate premature cell polarisation, and other factors are required to polarise Par complex components.

**PKC controls actomyosin polarisation by local activation**. To determine if a local activation of PKC triggers local actomyosin activation, we wished to apply a photo-activatable system to achieve precise activation of PKC. To this end, we used the CRY2 and CIB1 protein system, in which blue light triggers the binding of CRY2 and CIB1 as a heterodimer[43, 44]. We fused CIB1 with Zsgreen to visualise its localisation and the K-Ras CAAX motif to provide a membrane localisation signal. CRY2 was fused with the kinase domain of PKCα (CRY2-PKC-KD) and with mCherry, to visualise its localisation (Fig. 5i). We found that within 10 min of illumination with blue light in a defined cell contact-free region, CRY2-PKC-KD was recruited to this region, indicative of the translocation of the PKC kinase domain to the cell membrane and the effective binding of CRY2 and CIB1 (Fig. 5j; Supplementary Fig. 6a, b and Supplementary Movie 5). The cortical accumulation of CRY2-PKC-KD led to an enrichment of F-actin and phosphorylated MRLC within but not outside of the illuminated area (Fig. 5k). In contrast, no phosphorylated MRLC could be detected on the cell cortex without blue-light illumination. Blue light was also insufficient to trigger phosphorylated MRLC upregulation in embryos expressing constructs without the PKC kinase domain (Supplementary Fig. 6c, d). Together these multiple control conditions suggest that the upregulation of actomyosin recruitment was caused by PKC activation rather than any other non-specific effect. In agreement with our results above, the localisation of Pard6 remained unaffected. Together, these results show that local activation of PKC directly leads to a local activation of actomyosin but not Par complex polarisation.

**Inhibition of PLC–PKC abolishes MRLC apical localisation**. Since the above results demonstrated that PKC mediates local activation of actomyosin, we next wished to uncover the mechanism behind it. To this end, we first tested whether PLC–PKC inhibition leads to a decrease in the level of cortical myosin. We found that inhibition of PLC–PKC signalling, either by overexpression of PLC-DN or by inhibition of PKC, abrogated the accumulation of GFP-MRLC at the cell-contact-free cortex (Fig. 6a–f). It has been shown that MRLC has to be assembled into the myosin II complex to be recruited to the cortex and its phosphorylation is a pre-requisite for myosin II assembly[45, 46]. In agreement, we found that MLCK inhibition significantly reduced MRLC accumulation to the cell cortex (Fig. 6g, h). In light of

**Fig. 3** PLC–PKC activity is essential for polarisation at the 8-cell stage. **a** Scheme of the inhibitor (sphingosine, calphostin C) treatment. (**b**) DMSO, sphingosine and calphostin C-treated embryos immunostained for F-actin and Pard6. **c, d** Cell-contact-free enrichment of F-actin (**c**) or Pard6 (**d**) as a function of the IEA in embryos from (**b**). Each dot represents an individual measurement. **e** Percentage of polarised embryos in the groups of (**b**). The number of embryos per category is indicated. Embryos that did not show any blastomere with the enrichment of F-actin and Pard6 to the cell-contact-free surface were considered non-polarised. Otherwise embryos were classified as polarised. Data are shown as a contingency table. ****$p < 0.0001$, Fisher's exact test. **f** Quantification of polarisation in embryos from (**b**). Data are shown as mean ± s.d. ****$p < 0.0001$, unpaired two-tailed Student's $t$ test (three independent experiments). **g** Scheme of U73122 treatment and OAG rescue experiment. **h** DMSO, U73122 and U73122 + OAG-treated embryos immunostained for F-actin and Pard6. **i, j** Cell-contact-free enrichment of F-actin (**i**) or Pard6 (**j**) as a function of the IEA in embryos from panel h. Each dot represents an individual measurement. **k** Percentage of polarised embryos in the groups of (**h**). The number of embryos per category is indicated. Data are shown as a contingency table. ****$p < 0.0001$, NS = not significantly different, Fisher's exact test. **l** Quantification of polarisation in embryos from (**h**). Data are shown as mean ± s.d. ***$p < 0.001$, ****$p < 0.0001$, NS = not significant different, Kruskal–Wallis test and Dunn's multiple comparisons test (two independent experiments). **m** Scheme of the PLC-DN overexpression and OAG rescue experiment. **n** Ruby and PLC-DN overexpressing embryos ± OAG immunostained for F-actin and Pard6. **o** Percentage of polarised embryos in the groups of (**n**). The number of embryos per category is indicated. Data are shown as a contingency table. ****$p < 0.0001$, **$p < 0.01$, Fisher's exact test. **p** Quantification of polarisation in embryos from (**n**). Data are shown as mean ± s.d. **** $p < 0.0001$, **$p < 0.01$, *$p < 0.05$, Kruskal–Wallis and Dunn's multiple comparisons tests (two independent experiments). All scale bars, 15 μm

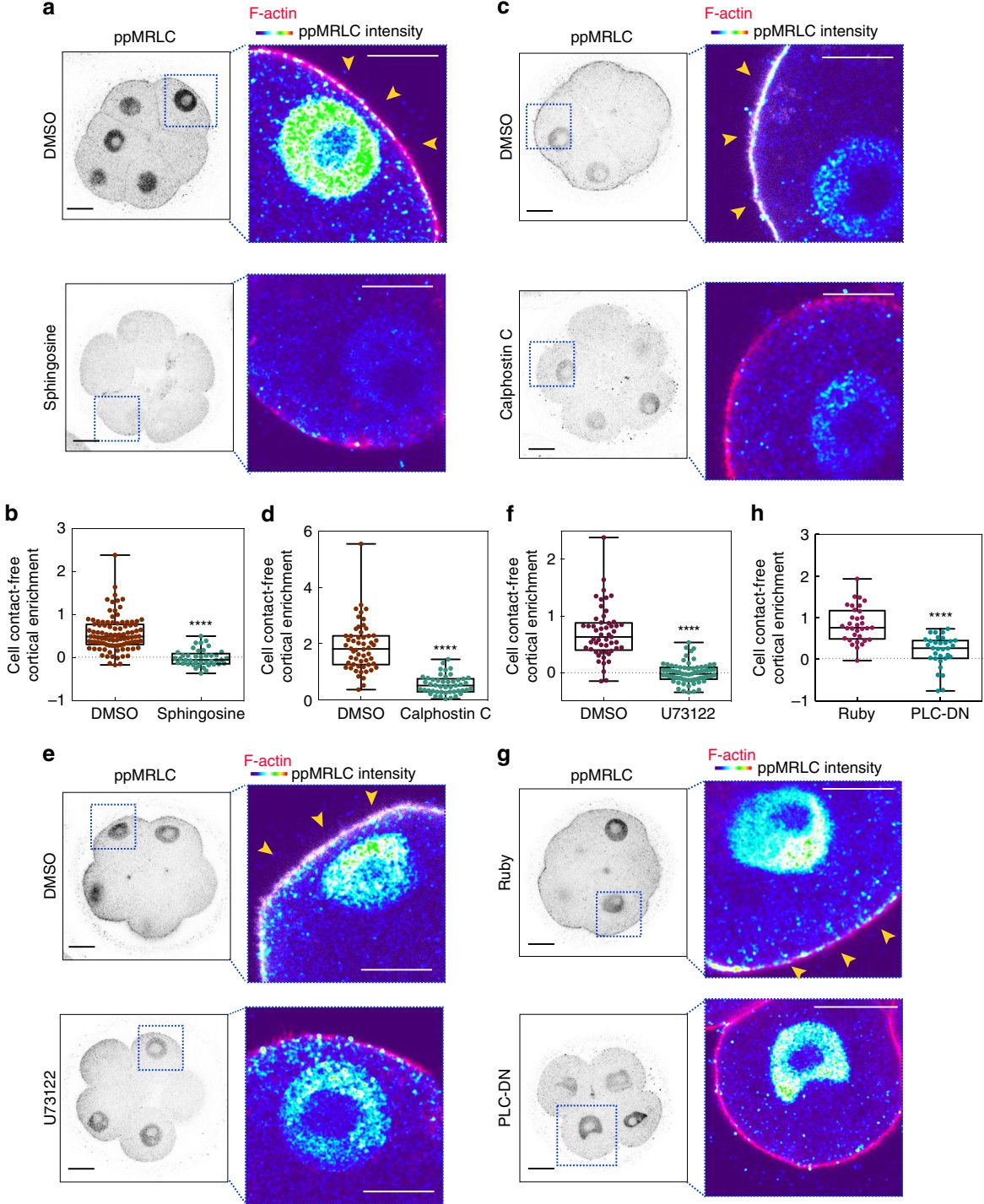

**Fig. 4** PLC–PKC inhibition abolishes apical ppMRLC accumulation. **a** DMSO and sphingosine-treated embryos stained for F-actin and ppMRLC (**b**). Quantification of ppMRLC enrichment at the cell-contact-free domain in embryos from (**a**) $N = 9$–10 embryos, two independent experiments. **c** DMSO and calphostin C-treated embryos stained for F-actin and ppMRLC. **d** Quantification of ppMRLC enrichment at the cell-contact-free domain in embryos from (**c**). $N = 11$ embryos for DMSO and $N = 9$ embryos for calphostin C, two independent experiments. **e** DMSO and U73122-treated embryos stained for F-actin and ppMRLC. **f** Quantification of ppMRLC cell-contact-free cortical enrichment in embryos from (**e**). $N = 20$ embryos for DMSO and $N = 22$ embryos for U73122, three independent experiments. **g** Ruby and PLC-DN-overexpressing embryos stained for F-actin and ppMRLC. **h** Quantification of ppMRLC cell-contact-free cortical enrichment enrichment in embryos from (**g**). $N = 7$ embryos for Ruby and $N = 8$ embryos for PLC-DN, two independent experiments. ****$p < 0.0001$, Mann–Whitney test. For all panels, arrowheads indicate apically localised ppMRLC and squares indicate the magnified regions. Data are shown as individual data points with Box and Whiskers graph (bottom: 25%; top: 75%; line: median; whiskers: min to max). All scale bars, 15 μm

these findings, we hypothesised that the reduction of MRLC cortical accumulation observed upon PLC–PKC inhibition might be caused by a reduction of MRLC di-phosphorylation. To address this possibility, we constructed a phosphomimetic mutant of GFP-MRLC (GFP-MRLC-DD), in which the two MLCK phosphorylation sites (T18, S19) were mutated to aspartic acid

(D) residues to mimic the di-phosphorylated conformation as this mutant has been shown to bypass the phosphorylation step required for the assembly of myosin filaments[47], although it is inefficient in recapitulating all of the functions of MRLC[48]. In line with these observations, we found that GFP-MRLC-DD could still accumulate at the cortex despite MLCK inhibition (Fig. 6h, i). In

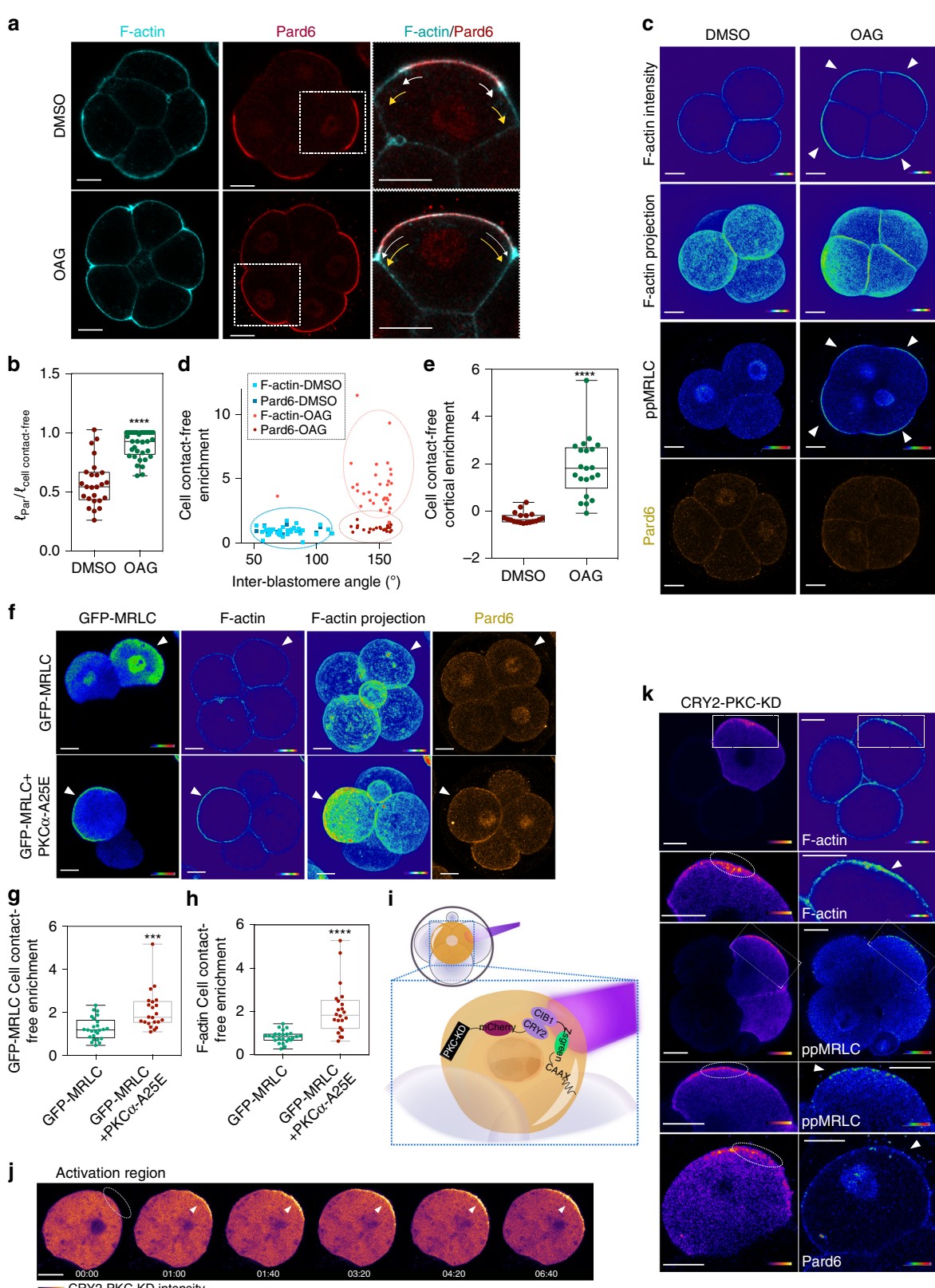

contrast, however, this mutant of GFP-MRLC was unable to gain cortical localisation following overexpression of PLC-DN or treatment with U73122 or sphingosine (Fig. 6a–f). Together, these results indicate that PLC–PKC signalling regulates the recruitment of MRLC to the apical cortex independently of MLCK-mediated MRLC phosphorylation, and that this apical localisation is necessary for the establishment of cell polarity at the 8-cell stage.

**RhoA regulates MRLC apical localisation downstream of PKC–PLC.** The above findings indicated that PKC activity controls the cortical recruitment of MRLC independently of MLCK-mediated MRLC phosphorylation and that both signalling pathways are required for de novo polarisation of the embryo at the 8-cell stage. However, the mechanism through which PKC controls MRLC localisation in the mouse embryo still remained unknown. We hypothesised that Rho proteins (Rho GTPases RhoA/B/C) might be potential candidates since they are involved in 8-cell stage polarisation, although the mechanism through which they might act has remained undetermined[49, 50]. To address if Rho proteins regulate MRLC localisation, we injected GFP-MRLC mRNA at the 4-cell stage, treated embryos with the Rho proteins inhibitor C3-transferase[51] at the 4- to 8-cell stage transition and examined the localisation of MRLC and Pard6 at the late 8-cell stage. All embryos in the control group had polarised GFP-MRLC and Pard6, while C3-transferase treatment strongly suppressed GFP-MRLC and Pard6 recruitment to the cell-contact-free domain (Supplementary Fig. 7a–d). To test whether Rho GTPases are the downstream effector of PLC–PKC signalling during cell polarisation, we used Forster resonance energy transfer (FRET)-based RhoA sensor (Raichu-RhoA-CR)[52]. We found that in the DMSO-treated control embryos, RhoA activity was gradually enriched in the cell-contact-free domain during compaction (Supplementary Fig. 7e, f), but U73122 treatment strongly abolished its polarisation to the cell-contact-free surface (Supplementary Fig. 7e, f), suggesting that PLC–PKC activity is responsible for polarising RhoA activity to the apical domain. To determine whether RhoA is the major component downstream of PLC–PKC to establish cell polarity, we wished to examine whether constitutively activate RhoA (RhoA-Q63L) could rescue the polarity defect caused by PLC–PKC inhibition. To this end, PLC-DN mRNA was injected at the 2-cell stage to block PIP$_2$ hydrolysis and, at the 4-cell stage, two blastomeres were injected with constitutively active RhoA-Q63L mRNA and GFP-MRLC mRNA to follow myosin II localisation (Fig. 7a). In agreement with our experiments above, we found that overexpression of

PLC-DN inhibited polarisation of Pard6 and GFP-MRLC when compared to control embryos (Fig. 7b–d). Remarkably, blastomeres in which RhoA-Q63L was overexpressed regained asymmetrically localised Pard6 and MRLC (Fig. 7b–d). To confirm these results further, we used the PLC inhibitor U73122, as an alternative approach to inhibit PLC–PKC signalling. To this end, we first cultured 4-cell stage embryos in the presence of U73122 (or in DMSO as a control) and, at the mid to late 4-cell stage, injected two blastomeres with GFP-MRLC mRNA alone (control) or with GFP-MRLC and RhoA-Q63L mRNAs (experimental group) (Fig. 7e). In accord with the above PLC-DN rescue experiment, we found that Pard6 and MRLC regained their polarised cortical localisation in blastomeres overexpressing RhoA-Q63L (Fig. 7f–h).

Finally, to determine whether RhoA activation at the 4-cell stage could advance the timing of Pard6 polarisation, we constructed a controllable RhoA photoactivation system in which the CRY2 construct was fused with RhoA mutant Q63L-C190R. Such RhoA mutant remains constitutively active but loses the ability to translocate to the membrane by itself, and thus cannot activate downstream effectors[53]. We found that blue-light activation effectively upregulated cortical localisation of phosphorylated MRLC. This upregulation was not observed in a membrane region not exposed to light (Supplementary Fig. 6d), confirming the specificity of our system. However, blue-light-mediated activation of RhoA was insufficient to recruit Pard6, similar to our findings when PKC was activated at this stage (Supplementary Fig. 7g).

Taken all together, our results indicate that PLC–PKC signalling is involved in the initiation of cell polarity in the mouse embryo at the 8-cell stage by regulating cortical localisation of MRLC via RhoA activation (Fig. 7i).

## Discussion

Establishing cell polarity is a critical step in embryo development. In the mouse embryo, the establishment of cell polarity is a pre-requisite for the first-lineage segregation to set apart the pluripotent progenitor cells of the foetus from the progenitors of the placenta[12]. Here, we demonstrate that cell polarisation in the mouse embryo can be divided into two phases: an initiation phase and a maturation phase. In the initiation phase, actomyosin becomes polarised to the apical domain at the 8-cell stage laying the foundation for the polarisation of Par complex to the apical domain, excluding actomyosin and forming a mature apical cap in the maturation phase. We also demonstrate that PLC–PKC signalling is necessary and sufficient to trigger actomyosin polarisation through

**Fig. 5** Ectopic activation of PKC expands the apical domain and induces premature actomyosin polarisation. **a** 8-cell stage embryos treated with DMSO or OAG and immunostained for F-actin and Pard6. Yellow arrows indicate the borders of the cell-contact-free domain and white arrows indicate the borders of the Pard6 apical domain. **b** Quantification of the Pard6 surface coverage rate in embryos from (**a**). The surface coverage rate is calculated as the ratio between the length of Pard6 positive domain and the length of the cell-contact-free surface. Data are shown as individual data points with Box and Whiskers graph. ****$p < 0.0001$, two-tailed unpaired Student's $t$ test. $N = 8$ embryos for DMSO and $N = 6$ embryos for OAG, four independent experiments. **c** 4-cell stage embryos treated with DMSO or OAG and immunostained for F-actin, ppMRLC and Pard6. Arrowheads indicate apically polarised F-actin or ppMRLC. **d** Cell-contact-free enrichment of F-actin and Pard6 as a function of the IEA in embryos from (**c**). Dots represent individual measurements. **e** Quantification of cell-contact-free cortical enrichment of ppMRLC in embryos from (**c**). Data are shown as individual data points with a Box and Whiskers graph (bottom: 25%; top: 75%; line: median; whiskers: min to max). ****$p < 0.0001$, Mann–Whitney test. $N = 18$ embryos for DMSO and $N = 10$ embryos for OAG, six independent experiments. **f** 4-cell stage embryos expressing GFP-MRLC or GFP-MRLC + PKCα-A25E in two blastomeres were immunostained for GFP, F-actin and Pard6. Arrowheads indicate the injected blastomeres. **g, h** Quantification of cell-contact-free enrichment of GFP-MRLC (**g**) or F-actin (**h**) in embryos from (**f**). Data are shown as individual data points with Box and Whiskers graph (bottom: 25%; top: 75%; line: median; whiskers: min to max). ***$p < 0.001$, ****$p < 0.0001$, Mann–Whitney test. $N = 6$ embryos, three independent experiments. **i** Scheme of the PKC-localised activation using the CRY2-CIB1 photoactivatable system. **j** Time-lapse snapshots of the localisation of CRY2-PKC-KD under localised blue-light illumination. **k** Blastomeres expressing CIB1-Zsgreen-CAAX and CRY2-PKC-KD were regionally illuminated using a 458 nm wavelength and immunostained for F-actin, ppMRLC and Pard6. Arrowheads or dotted circles indicate the illuminated region. Squares indicate the magnified regions. ($N = 24$ embryos, 11 independent experiments). All scale bars, 15 μm

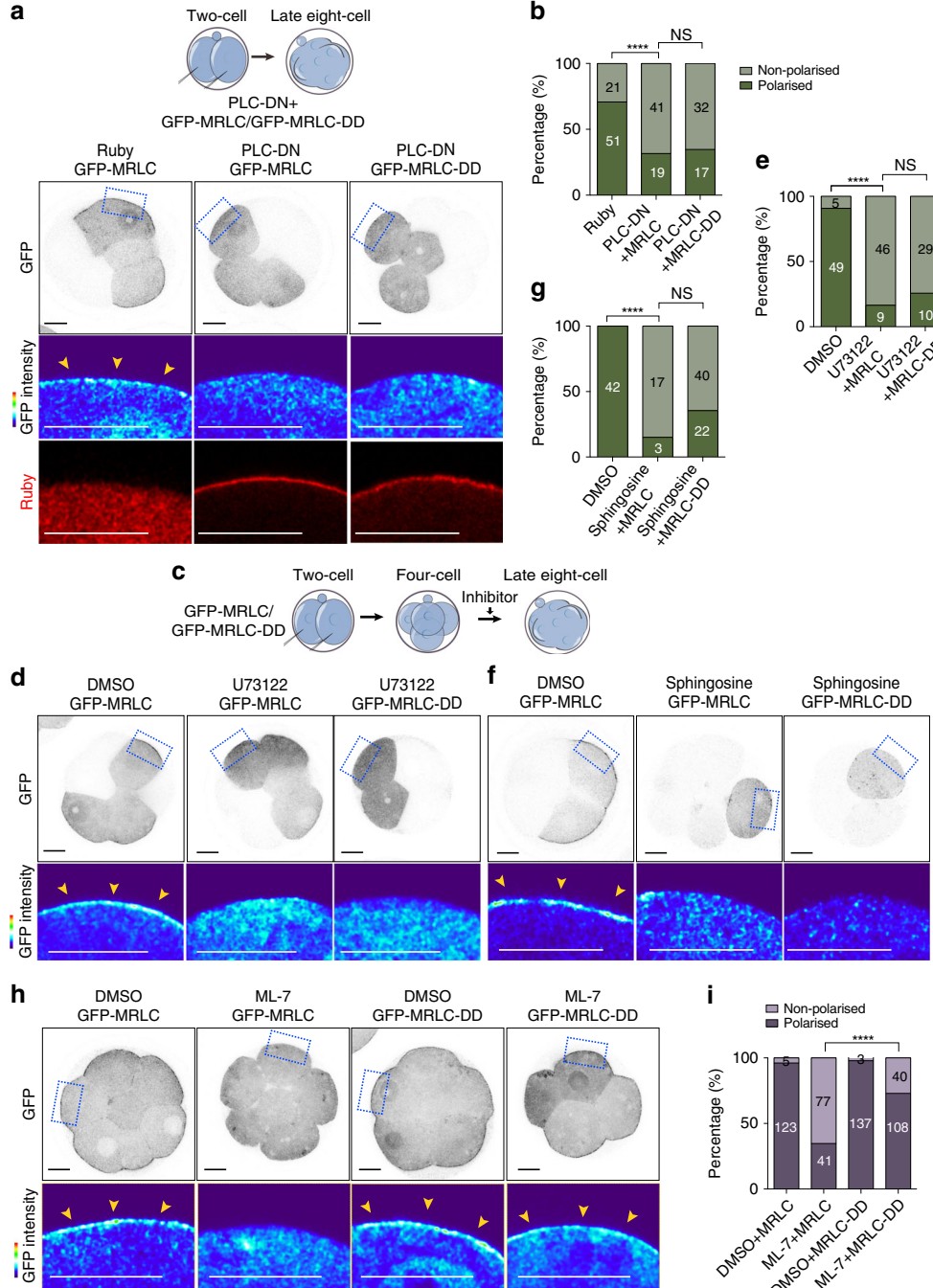

**Fig. 6** PLC–PKC inhibition prevents myosin II apical recruitment independently of myosin phosphorylation. **a** GFP-MRLC or GFP-MRLC-DD overexpressing embryos were injected with Ruby or PLC-DN and fixed at the late 8-cell stage. **b** Percentage of blastomeres showing a polarised GFP-MRLC or GFP-MRLC-DD in the groups of (**a**). $N = 17$ embryos for Ruby + MRLC, $N = 17$ embryos for PLC-DN + MRLC and $N = 17$ embryos for PLC-DN + MRLC-DD, three independent experiments. **c** Scheme of PLC–PKC inhibitor treatment experiments. **d** GFP-MRLC or GFP-MRLC-DD overexpressing embryos were treated with DMSO or U73122 and fixed at the late 8-cell stage. **e** Percentage of blastomeres showing a polarised GFP-MRLC or GFP-MRLC-DD in the groups of (**d**). $N = 11$ embryos for DMSO + MRLC, $N = 11$ embryos for U73122 + MRLC and $N = 9$ embryos for U73122 + MRLC-DD, four independent experiments. **f** GFP-MRLC or GFP-MRLC-DD overexpressing embryos were treated with DMSO or sphingosine and fixed at the late 8-cell stage. **g** Percentage of blastomeres showing a polarised GFP-MRLC or GFP-MRLC-DD in the groups of (**f**). $N = 20$ embryos for DMSO + MRLC, $N = 9$ embryos for sphingosine + MRLC and $N = 20$ embryos for sphingosine + MRLC-DD, two independent experiments. **h** GFP-MRLC or GFP-MRLC-DD overexpressing embryos were treated with DMSO or ML-7 and fixed at the late 8-cell stage. **i** Percentage of polarised blastomeres in the groups of (**h**).
$N = 18$ embryos for DMSO + MRLC, $N = 17$ embryos for ML-7 + MRLC, $N = 15$ embryos for DMSO + MRLC-DD, $N = 18$ embryos for ML-7 + MRLC-DD, two independent experiments. For all, the quantifications data are shown as a contingency table and the $n$ number in each bar indicates the total number of blastomeres analysed. Squares indicate the magnified regions. ****$p < 0.0001$, NS = not significantly different, Fisher's exact test. All scale bars, 15 μm

the Rho-mediated recruitment of myosin II to the apical cortex. Together, this reveals the molecular circuitry that triggers de novo polarisation of the mouse embryo at the 8-cell stage.

What initiates mouse embryo polarisation at the 8-cell stage has remained unknown. It has been suggested that this might result from changes in cell adhesion because the adherens junction component E-cadherin antagonises actomyosin at the

basolateral domain, acting as a negative regulator of apical domain formation[54]. However, other lines of evidence have suggested that cell–cell contacts might be insufficient to trigger cell polarisation because, for example, chimeras between 8-cell stage and earlier stage blastomeres show cellular polarisation exclusively in the 8-cell stage blastomeres[55]. In addition, it has been shown that the apical domain could be established in the

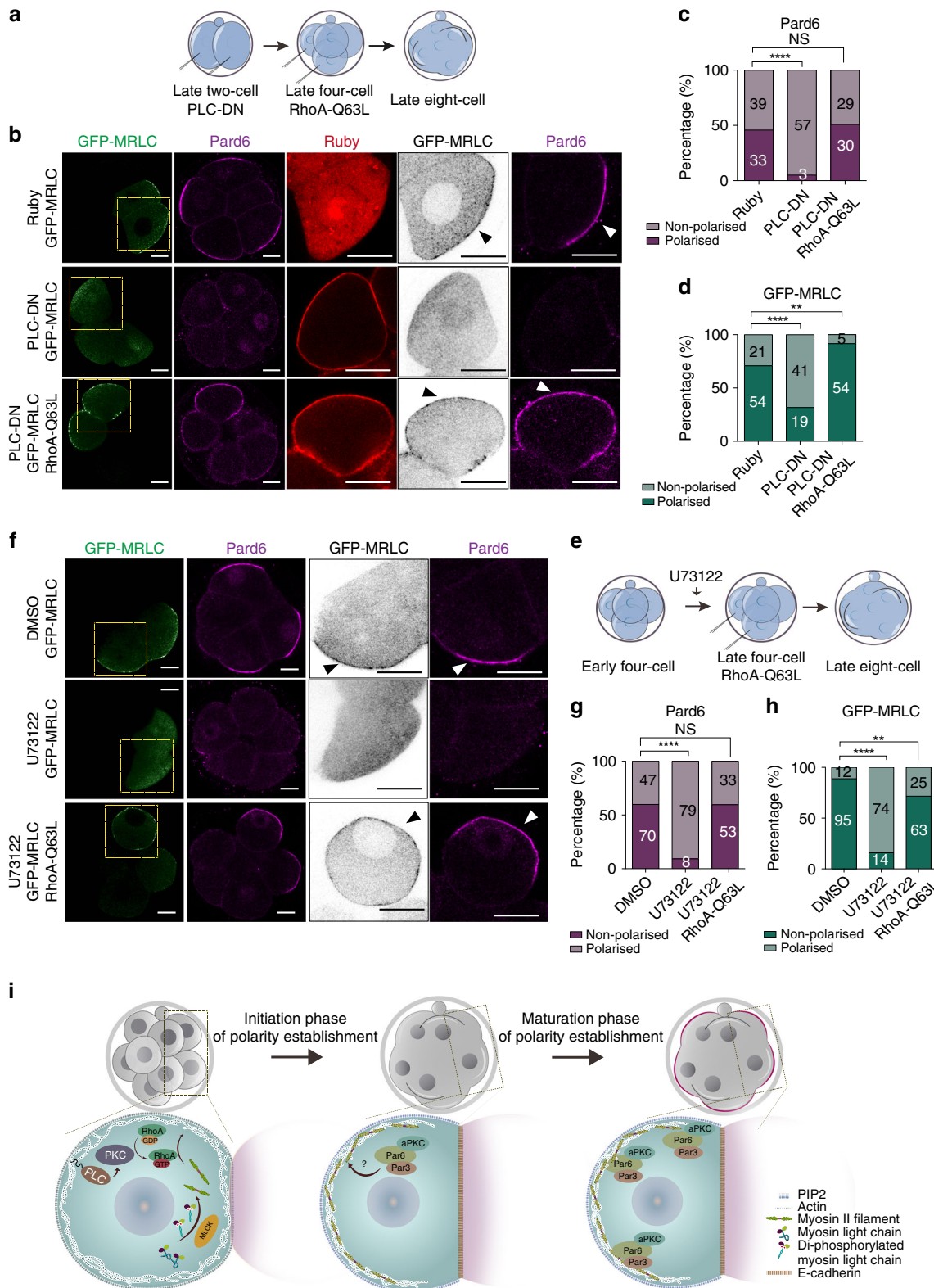

absence of E-cadherin or cell–cell contacts[12, 56] and that microtubules might be able to induce apical domain formation in a small proportion of cells independent of cell contact as a back-up mechanism[57]. Similarly, as polarisation happens in parallel to compaction, it has been postulated that compaction is required for apical domain assembly. Here, we show that compaction and polarisation are independent events as compaction can be specifically interfered without affecting polarisation, and vice versa. Therefore, the polarity phenotype caused by inhibition of PLC–PKC, MLCK or RhoA is specific to apical domain assembly, rather than a secondary effect of affecting compaction. The results we present here strongly indicate that the accumulation of actomyosin at the apical cortex of blastomeres at the early 8-cell stage is essential to set up cell polarity enabling the apical localisation of the Par complex. Our results also provide evidence that PKC is both necessary and sufficient to induce this asymmetric localisation of the actomyosin network as the first manifestation of cellular asymmetry. The early mouse embryo has multiple isoforms of PKC provided by maternally expressed genes, and therefore to eliminate PKC function genetically would require a complex combination of maternal and zygotic knockouts of multiple genes. To overcome this difficulty, we have applied a combination of specific pharmacological inhibitors, of either PKC or its activator PLC and dominant negative mutant constructs to eliminate PLC–PKC function specifically. Moreover, we could activate PKC by application of OAG, which broadened the domain of PKC activity and hence actomyosin activation, or by using an optogenetic approach to activate PKC and actomyosin locally. Application of these multiple approaches has allowed us to reveal that PKC is a rate-limiting factor for actomyosin polarisation, and thus PKC activity acts as a trigger to initiate polarisation at the 8-cell stage.

But how does PKC induce an asymmetric localisation of actomyosin? It is known that myosin is activated via phosphorylation, which is needed for its assembly into myosin filaments and subsequent cortical recruitment[46]. To generate a stabilised cortical myosin network, the actin cytoskeleton also needs to be assembled into a cortical actin meshwork, which is mediated by a number of actin cross-linking proteins, such as formins and Arp2/3[58]. In the muscle, PKC is known for its ability to control myosin localisation via phosphorylation[59]. Our results indicate that in the early pre-implantation mouse embryo PKC regulates myosin localisation through the activity of Rho GTPases. As RhoA is a common activator of formins, one possibility is that PKC controls myosin II cortical recruitment through formin-mediated actin reorganisation. It will be of future interest to reveal the dynamics of the apical actin meshwork and actin cross-linkers, and whether they are controlled by PKC–Rho signalling in the cleavage stage mouse embryo.

Our results identifying the asymmetric localisation of the actomyosin network as the initial step for mouse embryo polarisation echo the importance of the cortical actomyosin flow to localise anterior-determining proteins in the *Caenorhabditis elegans* zygote[60]. However, our results also reveal two key differences in the relationship between actomyosin and Par complex polarity between these two model systems. First, we find that upon polarisation of Par complex components, actomyosin becomes locally reduced and as a result forms a ring structure. This indicates a negative regulation of actomyosin by Par proteins, in agreement with the failure of actomyosin to form a ring structure upon elimination of aPKC[61]. Second, although we show that an apical contractile myosin meshwork is absolutely required for apical localisation of the Par complex in the mouse embryo, just as in the *C. elegans* embryo, it is not by itself sufficient. Indeed, we find that causing premature actomyosin polarisation at the 4-cell stage cannot, on its own, trigger apical enrichment of Par proteins. Thus, polarisation of actomyosin and Par complex are not necessarily coupled. It is likely that the intact actomyosin meshwork scaffolds additional factors, which in turn, possibly through physical binding, allow the Par complex components to accumulate on the actomyosin-enriched cell-contact-free surface. The identification of such additional factors will present a future challenge. Together, our results reveal how actomyosin polarisation breaks symmetry in the mouse embryo to regulate the de novo establishment of cell polarity in a temporally controlled manner.

## Methods

**Animals.** This research has been regulated under the Animals (Scientific Procedures) Act 1986 Amendment Regulations 2012 following ethical review by the University of Cambridge Animal Welfare and Ethical Review Body (AWERB). For the collection of embryos, F1 (C57BI6xCBA) females were superovulated by injecting 7.5 IU of pregnant mares' serum gonadotropin (PMSG; Intervet), followed by the injection of 7.5 IU of human chorionic gonadotropin (HCG; Intervet) after 48 h. The F1 females were mated with F1 males.

**Embryo culture and inhibitor treatments.** Embryos were recovered at the 2-cell stage in M2 medium and transferred to KSOM medium for culture and manipulations, as described previously[62]. Inhibitors were diluted in KSOM medium and applied as indicated in each experiment. For controls, the same amount of vehicle (DMSO or water) was used. Inhibitor concentrations and suppliers are as follows: D-erythro-sphingosine C-18 (2.5 µM; Caymanchem; in DMSO); Calphostin C (200 nM; Santa Cruz Biotechnology; in DMSO); U73122 (4.5–7.5 µM; Caymanchem; in DMSO); blebbistatin (25 µM; Sigma-Aldrich; in DMSO); ML-7 (20 µM; Sigma-Aldrich; in DMSO); Y-27632 (20 µM; Stemcell Technologies); C3-transferase (7 µg/ml; Cytoskeleton; in distilled water; the zona pellucida of 4–8-cell embryos was removed as described; 1-oleoyl-2-acetyl-sn-glycerol (OAG) (200 µM; Caymanchem; in DMSO), Latrunculin B (15 µM; Abcam; in DMSO).

**Microinjection.** The pRN3P vector was used as a backbone as previously described[63]. The mRNA was injected at the stage indicated at the following concentrations: PLCδ1-PH-Ruby (1 µg/µl); Ruby (900 g/µl); GFP-MRLC (600 ng/µl); PKCα-A25E (200 ng/µl); CIBN-Zsgreen-CAAX (400 ng/µl); CIBN-Zsgreen (400 ng/µl); CRY2-mCherry-PKC-KD (500 ng/µl); GFP-Myl12b-T18DS19D (600 ng/µl); Raichu-RhoA-CR (600 ng/µl); Clover (200 ng/µl); mRuby2 (200 ng/µl); RhoA-Q63L (150 ng/µl); CRY2-mCherry-RhoA-Q63L-C190R (600 ng/µl);

**Fig. 7** RhoA mediates MRLC and Pard6 cortical polarisation downstream of PLC–PKC signalling. **a** Scheme of the PLC–PKC rescue experiment using constitutively active Rho (RhoA-Q63L). **b** GFP-MRLC overexpressing embryos were injected with Ruby or PLC-DN and immunostained for GFP and Pard6 at the 8-cell stage. Arrowheads indicate the apical domain. **c**, **d** Percentage of polarised blastomeres based on either Pard6 (**c**) or GFP-MRLC (**d**) localisation in the groups of (**b**). N = 17 embryos for Ruby, N = 17 embryos for PLC-DN, N = 18 embryos for PLC-DN + RhoA-Q63L, two independent experiments. **e** Scheme of the PLC–PKC rescue experiment using constitutively active Rho (RhoA-Q63L). **f** GFP-MRLC overexpressing embryos were treated with DMSO or U73122 and immunostained for GFP and Pard6 at the 8-cell stage. Arrowheads indicate the apical domain. **g**, **h** Percentage of polarised blastomeres based on either Pard6 (**g**) or GFP-MRLC (**h**) localisation in the groups of (**f**). N = 21 embryos for DMSO, N = 17 embryos for U73122, N = 27 embryos for U73122 + RhoA-Q63L. **i** Summary model of the signalling events that trigger symmetry breaking and polarisation at the 8-cell stage. PLC-mediated PIP₂ hydrolysis activates PKC to trigger the apical enrichment of myosin II, a subsequent reorganisation of the cortical cytoskeleton and symmetry breaking. The apical polarisation of myosin II requires both RhoA activation (downstream of PLC–PKC signalling) and MLCK-mediated MRLC di-phosphorylation. At the late 8-cell stage, the actin-myosin enriched apical cortex recruits Pard6 (directly or indirectly) to the apical domain. For all the quantifications, data are shown as a contingency table and the *n* number in each bar indicates the total number of blastomeres analysed. ****p < 0.0001, NS = not significantly different, Fisher's exact test. Squares indicate the magnified regions. All scale bars, 15 µm

CRY2-mCherry (500 ng/μl); Ezrin-Ruby (400 ng/μl); LifeAct-eGFP (300 ng/μl). Microinjection was performed as previously described[63]. Briefly, embryos were placed in a drop of M2 medium on a depression glass slide, covered by paraffin oil. The microinjection was performed with Eppendorf Femtojet microinjector. Negative capacitance was used to facilitate the membrane penetration of nucleic acid. After injection, embryos were transferred to KSOM.

**Constructs preparation**. PLCδ1-PH-Ruby. The PH domain of Plcδ1 was PCR-amplified from pBSK-Plcd1-PH-eGFP[40] and cloned into the pRN3P vector.

pRN3P-Ruby and GFP-Myl12b. These constructs have been previously described[64].

GFP-Myl12b-T18DS19D: This construct was created by site-directed mutagenesis using GFP-Myl12b as template.

RhoA-Q63L: RhoA-Q63L was PCR-amplified from pRK5myc-RhoA-L63 (a kind gift of Mirna Perez-Moreno[65]) and inserted into pRN3P vector.

CIBN-Zsgreen and CIBN-Zsgreen-CAAX: The CIBN domain was PCR-amplified from pCMV-CIB1-mCerulean-MP (gift of Won Do Heo (Addgene plasmid #58366)). The Zsgreen was subcloned from pZsgreen-C1 (Clontech) and inserted downstream of CIBN. The CAAX motif of K-Ras was PCR-amplified from mouse liver complementary DNA (cDNA), and inserted downstream of Zsgreen.

CRY2-mCherry-PKC-KD: CRY2 domain and the mCherry coding sequence were subcloned from pCMV-CRY2-mCherry (gift of Won Do Heo (Addgene plasmid #58368)) into pRN3P; the kinase domain of PKCα (337-673aa) was PCR-amplified from pMTH-PKCalpha (gift of Frederic Mushinski (Addgene plasmid #8409)) and inserted downstream of CRY2-mCherry. PKCα-A25E: the coding sequence of Prkca was PCR-amplified from pMTH-PKCalpha and cloned into pRN3P. The A25E mutation was generated by site-directed mutagenesis.

Raichu-RhoA-CR: Raichu-RhoA-CR was subcloned from pCAGGS-Raichu-RhoA-CR (gift from Michael Lin (Addgene plasmid #40258)) and inserted into pRN3P.

Clover and mRuby2: Clover or mRuby2 coding sequences were subcloned from pCAGGS-Raichu-RhoA-CR and inserted into pRN3P vector.

CRY2-mCherry-RhoA-Q63L-C190R: RhoA-Q63L-C190R was generated by site-directed mutagenesis and inserted into pRN3p-CRY2-mCherry downstream of mCherry.

Ezrin-Ruby: Ruby was inserted into pRN3P, Ezrin was PCR-amplified from mouse adult kidney cDNA and inserted upstream of Ruby.

LifeAct-eGFP: LifeAct peptide coding sequence was cloned into pRN3P plasmid, eGFP coding sequence was subcloned and inserted downstream of LifeAct.

Primers for cloning and mutagenesis are in Supplementary Table 1.

**Immunofluorescence**. Embryos were fixed in 4% paraformaldehyde in PBS for 20 min at room temperature, and washed in PBST (0.1% Tween in PBS) three times. Embryos were permeabilised in 0.5% Triton X-100 in PBS for 20 min at room temperature, washed in PBST three times, transferred to blocking solution (3% bovine serum albumin) for 2 h and incubated with primary antibodies (diluted in blocking solution) at 4 °C overnight. After the incubation, embryos were washed in PBST and incubated with secondary antibodies (1:400 in blocking solution) for 1 h at room temperature. Embryos were stained with DAPI (1:1000 dilution, in PBST, Life Technologies, D3571) for 15 min, followed by two washes in PBST. Primary antibodies: rabbit polyclonal anti Pard6b (Santa Cruz, sc-67393, 1:200), rabbit polyclonal anti ppMRLC2 (Cell Signalling, 3674P, 1:200), mouse monoclonal anti PKCζ (Santa Cruz, sc-17781, 1:50), rabbit polyclonal anti CRB3 (Atlas antibodies, HPA013835, 1:50) and mouse monoclonal anti GFP (Nacalai Tesque Inc., 04404-84, 1:500). Secondary antibodies: Alexa Fluor 488 donkey anti-rat (Thermo Fisher Scientific, A21208), Alexa Fluor 568 donkey anti-rabbit (Life Technologies, A10042), Alex Fluor 647 donkey anti-rabbit (Life Technologies, A-21209) and Alexa Fluor 488 Phalloidin (Life Technologies, A12379).

**Imaging and data processing**. Imaging was done on a Leica-SP5 confocal using a Leica 1.4 NA 63X oil (HC PL APO) objective. Images were processed with Fiji software[66]. For the analysis of cortical fluorescence levels, a line with the width of 0.8 μm overlapping the entire cell-contact-free or cell-contact cortical area was defined based on the F-actin staining (as illustrated in Supplementary Fig. 1b); fluorescence levels were determined using the manager tool and plot profile function in Fiji; signals of different proteins were normalised against the average signal of the entire area for the plotting. Cortical signal enrichment was calculated as $(I_{cortex}-I_{cytoplasm})/I_{cytoplasm}$. Cell-contact-free enrichment was calculated as $I_{cell-contact-free}/I_{cell-contact}$. The length of cell-contact-free domain or Pard6 domain were measured using freehand line tool and manager tool in Fiji software.

The IEA between adjacent blastomeres was measured as described[11].

In live-imaging experiments, time-lapse frames were acquired every 20 min, using a 3–4 μm Z-step. Images were processed with Fiji software. LOWESS curves were calculated using R[67]. Non-linear regression curves were calculated using Prism software (http://www.graphpad.com).

**Optogenetics and regional PKC activation**. 2-cell stage embryos were injected with CIBN-Zsgreen-CAAX and CRY2-mCherry-PKC-KD or CRY2-mCherry-

RhoA-Q63L-C190R or CRY2-mCherry and cultured in the dark. After 3 h, embryos were transferred to an optical glass bottom culture dish (MatTek Corporation) mounted with M2 medium and imaged under Leica TSC SP5 confocal microscopy (×63 oil objective). Laser power of 458 nm wavelength (8% for CRY2-mCherry-PKC-KD and CRY2-mCherry, 4% for CRY2-mCherry-RhoA-Q63L-C190R) was used to illuminate a defined region in the cell-contact-free domain of the blastomeres using the region of interest tool in Leica Las AF software. The laser was illuminated for 5 s and followed with a 20 s interval. Each individual embryo was imaged for 15 min.

**RhoA-FRET imaging and data analysis**. The structure of RhoA FRET sensor Raichu-RhoA-CR was as described previously[68]. Raichu-RhoA-CR, Clover mRNA and mRuby2 mRNA were injected into 2-cell stage embryos, and at the 4–8 cell stage, embryos were imaged under Leica TSC SP8 confocal microscope (Leica fluotar VISIR 0.95NA ×25 water objective). Images were acquired using the following excitation and emission settings: clover: 488 nm excitation, 505–540 nm emission; FRET: 488 nm excitation, 580–650 nm emission; mRuby2: 568 nm excitation, 580–650 nm emission. Clover mRNA-injected embryos and mRuby2 mRNA-injected embryos were imaged first to determine the signal bleed-through (BT). Next, Raichu-RhoA-CR-expressing embryos were imaged with 20 min interval. For data analysis, the FRET and co-localisation analyser plugins in Fiji software were used to analyse the co-localisation between donor, FRET and acceptor channels to eliminate the noise generated by detectors. The radiometric image of FRET-co-localisation and donor images were generated using the image calculation function in Fiji. Cell-contact-free/cell-contact RhoA-FRET ratio is calculated by $I_{cell-contact-free}/I_{cell-contact}$.

**Statistics**. Statistical methods are indicated for every experiment in the corresponding figure legend. Qualitative data is presented as a contingency table and was analysed using Fisher's exact test. Normality of quantitative data was first analysed using D'Agostino's K-squared test. For data showing a normal distribution, unpaired two-tailed Student's $t$ test was used to analyse the statistical significance (two experimental groups). Differences in variances were taken into account performing a Welch's correction. For data that did not present a normal distribution, a Mann–Whitney U-test (two experimental groups) or a Kruskal–Wallis test with a Dunn's multiple comparison test (more than two experimental groups) were used to test statistical significance. Statistical analyses were performed using Prism software (http://www.graphpad.com).

**Data availability**. The authors claim that all relevant data of the findings in this work are provided within the paper and Supplementary Information files. Raw data are available from the corresponding author upon request.

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

## Acknowledgements

We thank all colleagues in our lab and D. Glover for the very helpful suggestions on the manuscript. This work was supported by the Wellcome Trust. M.Z.G. is a Wellcome Trust Senior Research Fellow, M.Z. is supported by the Cambridge Trust, and M.N.S. is an EMBO Postdoctoral Fellow. We would like to dedicate this work to M. Johnson, a pioneer of cell polarisation studies.

## Author contributions

M.Z. designed and conducted experiments, analysed and interpreted the data with the help of all co-authors. C.Y.L. performed pilot experiments. M.N.S. discussed and helped to interpret the data. M.Z.-G. conceived and supervised the project, and helped to interpret the data. The manuscript was written by M.Z., M.N.S. and M.Z.-G.

## Additional information

**Competing interests:** The authors declare no competing financial interests.

