## [Peer Review File · Nature Communications]

Reviewers' Comments:

Reviewer #1:

Remarks to the Author:

In this manuscript, Zhu, Zernicka-Goetz and colleagues reveal a mechanism explaining how cell polarity arises during early mammalian development. Their experiments and analyses are of high quality and their paper starts to solve an important question in the field of mammalian development, which should also be of interest to the wider developmental biology community.

The authors show that cell polarity in the early mouse embryo is built in two stages:

First, an apical actomyosin network is formed and reorganized. This requires PLC-PKC signaling, which polarizes actomyosin by Rho-dependent recruitment of apical myosin II. Then, the Par complex localizes to the apical domain, excluding actomyosin and forming a mature apical cap.

They have used a variety of clever manipulations. These include two independent approaches to perturb actomyosin in an acute (temporally controlled) manner, impressive experiments with a RhoA FRET sensor and the CRY/CIB optogenetic system. The combination of these approaches enabled them to tease apart molecular events and provide a detailed understanding of how cell polarization is first established in vivo. Moreover, they provide convincing analyses of myosin states using phospho-specific antibodies, which has remained difficult to achieve in the mouse embryo. And they further provide a detailed analysis of myosin phosphorylation from the 4- to late 8-cell stage.

Overall, the findings reported are of high interest and I support publication of this study. However, I suggest the authors address the following specific points:

The statement "Inhibition of both MLCK and myosin II ATPase also abolished compaction" should be followed by a reference to a figure.

"At the 16-cell stage, polarisation of the outer cells is responsible for activating Cdx2 expression". Please include a reference.

"This suggests that the upregulation of actomyosin activity". I suggest the authors refrain from stating 'activity' as this term applies well to myosin but not actin.

Fig. 1c & Fig. 1f: How are Phase I and II defined?

Is it time post-division?

What does the shading indicate?

Why is inter-blastomere angle abbreviated as IEA not IBA?

Fig. 4e & Fig. 4g: To exclude effects on the actomyosin deriving from cytoskeletal changes during division, the authors could show the ppMRLC intensity in cells where an interphase nucleus is visible.

The authors should keep the labeling of figures consistent – i.e., always Pard6 or Par6, but not both.

Given that they show the ATPase activity of Myosin II is not required for actomyosin polarization, it would be interesting to clarify what role they propose Myosin II has at the apical cortex?

The authors state that actomyosin (as demonstrated by F-actin and GFP-MRLC) is "excluded to the periphery of the Pard6 domain" at the late 8-cell stage. However, while there is clearly an accumulation of actomyosin in a ring surrounding the Pard6 domain, their fluorescence intensity

measurements in Fig. 1 and F-actin/Pard6 expressing embryos in Figures 1-3 suggest some overlap of Pard6 and actomyosin within the ring. Can the authors clarify this point?

Reviewer #2:

Remarks to the Author:

In this interesting manuscript, the authors characterise apical-basal polarisation of the early mouse embryo. A nicely performed set of imaging experiments reveal polarisation of F-actin with the classic Par6-dependent apical polarity machinery, which ultimately resolves into an apical ring. Overall, the results are solid and novel, and thus deserving of publication in Nature Communications.

Minor comments:

It would be interesting, if technically feasible, to examine the localisation of some of the other apical determinants during this process. For example Par3, Crb3, PALS1, aPKC ζ . It would also be interesting to examine FERM domain proteins as these may link the apical determinants to F-actin cytoskeleton.

Reviewer #3:

Remarks to the Author:

Zhu and colleagues examine the role of the PKC signaling pathway in initiating polarity in mouse embryos. They first show, through staging of fixed embryos, that apical accumulation of F-actin and non-muscle myosin precedes the apical accumulation of Pard6. Through use of drugs and elegant photoactivatable proteins, they identify PKC signaling, through RhoA, as a trigger that induces apical actomyosin. Interestingly, apical actomyosin is required for Pard6 accumulation but is not sufficient, indicating the presence of another trigger that allows Pard6 to localize in an actomyosin-dependent manner.

This paper connects several dots to reveal a more coherent pathway for inducing polarity, although several important questions are left unanswered. I found the data high-quality and quantitative, and I agree with most of the interpretations save some as noted below. How polarity is triggered in mammalian embryos is a question of considerable interest to developmental biologists and those interested more generally in cell polarization, so the findings presented should have broad appeal. One point that needs further investigation is the role of myosin (and actin) in Pard6 localization, and I feel that the authors should look at additional markers for Par protein localization. I have several additional minor suggestions for improving the manuscript.

Major comments

1. The negative blebbistatin results are worrisome to me, given that findings are different with inhibitors of myosin activity. How do the authors reconcile these results? One possibility is that inhibitors have varying degrees of effectiveness. Experiments should be performed with all three inhibitors to determine how effectively they block cytokinesis from the 4-8 cell division (I do see evidence of at least some multinucleate cells). Also, experiments should be performed to determine whether F-actin is required for Pard6 apical enrichment.
2. Basing Par protein behavior on one marker seems problematic, especially considering that there are multiple paralogues of each of the par genes and Par proteins do not always colocalize. aPKC and Par3 should also be examined in key experiments (timing of apical enrichment, dependence on PKC).

Minor comments

1. Abstract – it was not demonstrated that the Par complex excludes actomyosin to the edge of the apical domain. Language should be softened to suggest that this is the case.

2. Introduction – “In contrast to the development of embryos of the majority of metazoan species, mammalian embryos acquire cell polarity de novo at a species-specific developmental stage.” I disagree with the first part of this sentence; I can think of many animals that acquire polarity at a discrete point after cleavage has begun.
3. P.3. What do actin and myosin look like at different stages of the 4-cell stage? In other words, could any of their dynamic behavior during the 8-cell stage simply reflect cell-cycle-dependent localization patterns?
4. P. 3, first paragraph. Omit “leading to the first cellular asymmetry during embryo compaction.” E-cadherin asymmetry occurs prior to the appearance of actin and myosin asymmetry.
5. What happens to activated myosin upon blebbistatin treatment?
6. I would not say that Y-27632 treatment “abolished cell polarization” since F-actin still appears polarized (Fig S1e). This needs to be noted in the text.
7. P. 5-6. It’s unclear why PLC inhibition does not prevent cleavage if it blocks myosin localization. What happens to actin and myosin during cleavage if PLC is inactivated?
8. Discussion – Several points in the discussion deserve additional attention. How do the authors think that actin and myosin are working to recruit PAR proteins? In *C. elegans*, actomyosin cortical flows carry PAR proteins to the anterior (at the same time), so this mechanism does not fit the authors’ data. Also, what might be the cue that induces PAR asymmetry independently of actomyosin? Finally, how might PKC activity be controlled?

Reviewer #4:

Remarks to the Author:

This is a detailed account of actomyosin symmetry-breaking within the early mammalian embryo, which is an important and underexplored topic with a high level of scientific interest.

There are some beautiful and interesting results shown in the paper but as it stands, there are some major issues that would need resolving before it is suitable for publication. If these issues can be resolved then the paper would be an excellent and important publication for Nature Communications.

I also suggest that the manuscript could be more concise by more grouping of PKC-related experiments for example. This would make it easier to follow and to understand the main conclusions.

Major points

1. The authors have not separated compaction and polarisation phenotypes:

E.g. figure 2a ML-7 treated cells: Given the massive difference in cell compaction (and therefore in the general orientation and geometry of the cells) I suggest that this data cannot be used to draw conclusions about the role of actomyosin on cell polarity. To conclude that “MLCK and ROCK-mediated phosphorylation of MRLC is required for actomyosin and Par polarization”, the authors would need robust control data demonstrating the alteration of actomyosin and Par polarisation without compaction defects. This issue has not been addressed. E.g. in figure 3h, U73122 causes compaction defects and polarity defects. Both types of defects are rescued by the addition of OAG, making it impossible to distinguish whether the polarity defects were caused by PLC inhibition itself or by compaction inhibition generally.

2. The definition of ‘apical’ and ‘basal’ is not clear:

This point is related to point 1. Since many of their manipulations affect cell compaction and therefore cell interaction and arrangement, I worry that the authors have not addressed how this might affect the axis of polarity within these cells. E.g. supplementary figure 6e: RhoA is still activated in a polarised manner but in a different subcellular location. I don’t think it is appropriate to define the contact-free edges of the cells as ‘apical’ in situations where compaction is

prevented. There are also examples where there appear to be centrally-located accumulations of 'apical' proteins (e.g. the late 8-cell-2 in figure 2g has a centrally located ring of ppMRLC and there is a centrally located accumulation of F-actin and Pard6 in Y-27632 treated cells in supplementary figure 1e). What counts as 'apical' in these cases?

The distinction between 'apical' and 'basal' is especially important given the author's analyses of intensity levels from these regions. If there is apical domain localisation at the centre of the embryo but this is being classed as 'basal' then this would confuse the results. The authors should clearly illustrate how they sample their data (e.g. show the ROIs used to measure mean intensity levels).

3. Line analyses are not always appropriate

Polarity proteins such as Pard6 often initially appear in 'spot'-like clusters, rather than being "distributed equally around the cell cortex". This appears to be the case here (see spots of Pard6 around the cell cortex in figure 1a). This makes a line analysis inappropriate and brings into question the result that actomyosin polarisation preceded Pard6 polarisation. Could the authors also analyse intensity around the circumference of cells in figure 1a. This would also aid in the analysis of 'actin ring formation', which the authors mention but which is not possible to see clearly in figure 1a (it's a bit clearer in the example shown in figure 5a but an analysis would still help).

Also, what determines which cell polarises first? Are the authors consistent with which cell they select for analysis? E.g. in figure 1d, Pard6 is clearly polarised by the mid-late 8-cell stage and on the second cell appears polarised in relation to background levels even at the mid 8-cell stage but this cell has not been analysed.

4. The sufficiency of actomyosin asymmetry to polarize Par complexes is not demonstrated:

To say that "asymmetry of the functional actomyosin network is required for Par complex polarity" (i.e. it is necessary) would require a robust compaction control (see point 1 above). Even so, to say that "induction of this cytoskeletal asymmetry is an absolute pre-requisite for the cortical enrichment of the Par polarity complex to establish cell polarity" would require the sufficiency of actomyosin asymmetry to mediate Par asymmetry. This has not been shown. However, some of the data that the authors say does not show an upregulation of Pard6 seems like it actually does. E.g. it looks to me like there is a small upregulation of Pard6 in the OAG treated embryo shown in figure 5c. This is quantified in supplementary figure 4C and is significantly different. Why do the authors say that there is no apical enrichment of Pard6? Similarly, in figure 5f, there is a polarised enrichment of Pard6 in the cells injected with PKCa-A25E. However, this enrichment is in opposite sides of the 2 injected cells and these cells have different levels of construct expression making it hard to interpret this data.

5. Optogenetic approaches need further analysis:

The author's optogenetic experiments allow them to directly determine the causative links in the molecular cascades. However, this data requires further explanation and analysis:

- figure 5i-k and video 5: why is CIBI-CAAX only localised to one part of the cell? Is it the specific ROI that denotes the subcellular specificity of PKC activation or is it due to the subcellular localisation of CIBI-CAAX and how was this achieved?

- the authors should explain why the recruitment of the optogenetic effectors (kinase domain of PKC and RhoA L63-C190R) to the membrane causes specific activation and how they have avoided background levels of activation. Also, how did they achieve reproducible levels/control for different levels of each of the heterodimerization components between experiments? E.g. Sup fig 6g: The levels of CRY2-RhoA-L63-C190R are very different between the 2 panels. How do the authors control for this and be sure that RhoA is not activated without cortical localisation? There is no description either in the figure legend or main text/methods for how RhoA is specifically activated using this method.

- Results should be quantified and the authors should show all 3 channels (e.g. in figure 5: CIBI-

CAAX, CRY2-PKC-KD and F-actin/ppMRLC/Pard6) levels before and after optical stimulation to control for background activation and for intrinsic differences in individual cell polarisation. Illumination ROIs should be clearly defined and made distinct from boxes denoting magnified regions (e.g. figure 5J and K).

- Internal control cells should also be quantified
- I'm not sure that the Cryptochrome reference used (Liu et al 2008) is the most appropriate in isolation. It would be a good idea to also include a paper that shows the use of these proteins for experimental heterodimerization. I believe the first example of this for the CRY2 system is Kennedy et al 2010 Nature Methods).

6. The authors do not show any data demonstrating that apical actomyosin initiates 'fate decisions'. This should therefore be removed from the title.

Minor points

- It is important for the authors to be specific about what they mean by 'polarity' throughout the paper. Eg. First line of page 9 "this apical localisation is necessary for the correct establishment of cell polarity at the 8-cell stage". The data in figure 6 does not address Par6 polarity so this is only referring to actomyosin polarity.
- The authors should define acronyms on first use: e.g. PLC
- Methods: For microinjection the total amount of RNA injected should be listed, not just the concentration of solution.
- In the 'statistics' section, first line: 'qualitative data' is the wrong terminology here since qualitative data cannot be statistically analysed.

Specific figure points

Figure 1

- In Figure 1a, panels A and B need labelling. Perhaps use i and ii to distinguish from figure 1a and b?
- To see the actin ring in figure 1a more clearly, the actin 3D channel should be shown alone
- I think there may be an issue with directly comparing the fluorescence levels of Pard6 and F-actin since levels of F-actin are higher from the start. Perhaps a better read-out would be the normalised fluorescence.
- Can the analyses in Figure 1c and 1f (as well as similar analyses throughout the paper) be statistically analysed?
- The white read out in the MRLC channel makes it hard to interpret the projection. This is the case throughout the paper.
- Figure 1g: colours are too similar – hard to distinguish GFP-MRLC from background of cell body. I like the upper panel but the lower panel doesn't accurately reflect the basolateral localisation of F-actin and GFP-MRLC so could be a little confusing. Perhaps it is not required?

Figure 2

- See major points 1 and 2 above
- Figure 2a: I cant see the Pard6 staining in the first and last panels, even though this was visible at similar stages in figure 1.
- Could the lack of effect of blebbistatin on polarisation be due to the later treatment?
- Figure 2f – should explain in legend how apical enrichment values are calculated. Same for figure 4b,d,f and h

Figure 3

- Supplementary videos 3 and 4 should be labelled.

Figure 5

- See major points 4 and 5 above

- Figure 5b – explain the y-axis somewhere obvious, like the figure legend.
- Figures 5c and 5f: The authors should clarify what embryonic stage these experiments occur at – 4 cell stage?
- The reference to supplementary figure 3d should be supplementary fig 4d.

Figure 6

- Supplementary fig 6e – see major point 2 above.
- I found this section quite hard to follow. A diagram of hypotheses and conclusions would help.

Reviewers' comments:

Reviewer #1 (Remarks to the Author):

In this manuscript, Zhu, Zernicka-Goetz and colleagues reveal a mechanism explaining how cell polarity arises during early mammalian development. Their experiments and analyses are of high quality and their paper starts to solve an important question in the field of mammalian development, which should also be of interest to the wider developmental biology community.

The authors show that cell polarity in the early mouse embryo is built in two stages:

First, an apical actomyosin network is formed and reorganized. This requires PLC-PKC signaling, which polarizes actomyosin by Rho-dependent recruitment of apical myosin II. Then, the Par complex localizes to the apical domain, excluding actomyosin and forming a mature apical cap.

They have used a variety of clever manipulations. These include two independent approaches to perturb actomyosin in an acute (temporally controlled) manner, impressive experiments with a RhoA FRET sensor and the CRY/CIB optogenetic system. The combination of these approaches enabled them to tease apart molecular events and provide a detailed understanding of how cell polarization is first established in vivo. Moreover, they provide convincing analyses of myosin states using phospho-specific antibodies, which has remained difficult to achieve in the mouse embryo. And they further provide a detailed analysis of myosin phosphorylation from the 4- to late 8-cell stage.

Overall, the findings reported are of high interest and I support publication of this study. However, I suggest the authors address the following specific points:

We thank the reviewer very much for his/her appreciation of the significance and quality of our work and all of his/her helpful comments. We have modified the text and the figures according to the reviewer's suggestions. Detailed point-to-point responses are as follows:

1. The statement "Inhibition of both MLCK and myosin II ATPase also abolished compaction" should be followed by a reference to a figure.

Following the reviewer's suggestion, we have added a reference to the figure (page 4, line 12; line 23).

2. "At the 16-cell stage, polarisation of the outer cells is responsible for activating Cdx2 expression". Please include a reference.

We have added two references to support this statement (page 6, line 22).

3. "This suggests that the upregulation of actomyosin activity". I suggest the authors refrain from stating 'activity' as this term applies well to myosin but not actin.

We agree with the reviewer that our results show upregulation of myosin activity but accumulation of actin. Therefore, we now indicate that PKC induces the accumulation of actomyosin (page 9 line 2).

4. Fig. 1c & Fig. 1f: How are Phase I and II defined?

Is it time post-division?

What does the shading indicate?

We apologise for not clarifying this clearly, we have now corrected it.

To answer reviewer's question: phase I and phase II are defined based on the status of actomyosin and Par complex polarisation. Specifically, phase I refers to the period when only actomyosin, but not the Par complex, has polarised to the cell-contact free domain; phase II refers to the period when actomyosin is already polarised at the cell-contact free surface and the Par complex gradually polarises to build a mature apical domain.

We added shading to help to show the tendency of the F-actin/Pard6 polarisation pattern in relation to compaction/8-cell stage progression. To accurately display the tendency, we have revised our illustration method as shown in revised Fig. 1c and Supplementary Fig. 1e. Briefly, we fitted the plots into a "locally weighted scatterplot" (LOWESS) curve to simulate the general tendency of the F-actin/Pard6 apical/basal signal ratio in relation to the inter-blastomere angle. The shadow around the line indicates 95% confidence interval.

5. Why is inter-blastomere angle abbreviated as IEA not IBA?

This is to differentiate inter-blastomere angle from intra-blastomere angle. Both types of angle have been as a readout of compaction^{1,2}. To better clarify our method of measurement and to avoid confusion in reading our graphical figures (as during compaction, the inter-blastomere angle increases whereas the intra-blastomere angle decreases), we adopted IEA as an abbreviation.

Fig. 4e & Fig. 4g: To exclude effects on the actomyosin deriving from cytoskeletal changes during division, the authors could show the ppMRLC intensity in cells where an interphase nucleus is visible.

We thank the reviewer for this very helpful suggestion. We agree that this will help to clarify our phenotype and therefore changed our representative images and now compare the levels of actomyosin in the plane where the interphase nucleus can be seen (modified Fig. 4).

The authors should keep the labeling of figures consistent – i.e., always Pard6 or Par6, but not both.

We thank the reviewer for the comment and apologise for the inconsistency in labelling. We have now revised all the labelling and names to be consistent.

Given that they show the ATPase activity of Myosin II is not required for actomyosin polarization, it would be interesting to clarify what role they propose Myosin II has at the apical cortex?

We agree with the reviewer that given that the ATPase activity of Myosin II is not required for polarisation it is important to discuss what is myosin II function. We have added the relevant discussion in the revised manuscript (please see page 13, line 8). Briefly, we propose that the actomyosin cytoskeleton provides positional information for Par complex cortical recruitment. However, Par complex recruitment requires additional proteins that may function as a scaffold between actomyosin and Par complex.

The authors state that actomyosin (as demonstrated by F-actin and GFP-MRLC) is “excluded to the periphery of the Pard6 domain” at the late 8-cell stage. However, while there is clearly an accumulation of actomyosin in a ring surrounding the Pard6 domain, their fluorescence intensity measurements in Fig. 1 and F-actin/Pard6 expressing embryos in Figures 1-3 suggest some overlap of Pard6 and actomyosin within the ring. Can the authors clarify this point?

We thank the reviewer for this comment. We agree that the exclusion of actomyosin by Par complex is progressive rather than immediate. The observed exclusion of actomyosin by the Par complex is supported by the observation that in aPKC maternal zygotic knockout embryos, actomyosin remains uniformly enriched at the apical domain and fails to form the actomyosin ring structure at the late 8-cell stage³. The temporal lagging of actomyosin exclusion by Par complex could be due to the complex post-transcriptional modifications such as protein phosphorylation, the level of which has been shown to be upregulated during the 8-cell stage polarisation process⁴; or proteasome mediated protein degradation of apical proteins at the basal site; or the involvement of other proteins.

Reviewer #2 (Remarks to the Author):

In this interesting manuscript, the authors characterise apical-basal polarisation of the early mouse embryo. A nicely performed set of imaging experiments reveal polarisation of F-actin with the classic Par6-dependent apical polarity machinery, which ultimately resolves into an apical ring. Overall, the results are solid and novel, and thus deserving of publication in Nature Communications.

Minor comments:

It would be interesting, if technically feasible, to examine the localisation of some of the other apical determinants during this process. For example Par3, Crb3, PALS1, aPKC ζ /zeta. It would also be interesting to examine FERM domain proteins as these may link the apical determinants to F-actin cytoskeleton.

We thank the reviewer for his/her kind remarks about our work and for the helpful suggestion. To address this point, we have also examined the localization pattern of Crb3 and PKC ζ at early (within 1hr post-cell division), mid (3-4 hrs post-cell division) and late 8-cell stages (5-8 hrs post-cell division), and compare their localisation pattern with actomyosin. Similar to our earlier result with Pard6, F-actin progressively polarised to the apical domain during the early to mid 8-cell stage, whereas Crb3 and PKC ζ remained cytoplasmic and unpolarised. Only at the late 8-cell stage when F-actin was finally polarised, PKC ζ and Crb3 were detected at the cell-contact free surface (please see new supplementary Fig. 1c,g).

To determine the localisation of FERM family proteins throughout the 8-cell stage we have performed live imaging of embryos expressing Ezrin-RFP (FERM domain proteins) and Lifeact-GFP (live marker for actomyosin). We observed that in agreement with our immunostaining results, during compaction, Lifeact-GFP polarised to the cell-contact free surface. Only after Lifeact-GFP had polarised, did Ezrin-RFP begin polarising apically to localise inside the actomyosin ring structure (please see new supplementary Fig. 1h,i). Together, these results indicate that the actomyosin network polarises apically earlier than Par complex proteins and therefore can be considered as a symmetry breaking event.

Reviewer #3 (Remarks to the Author):

Zhu and colleagues examine the role of the PKC signaling pathway in initiating polarity in mouse embryos. They first show, through staging of fixed embryos, that apical accumulation of F-actin and non-muscle myosin precedes the apical accumulation of Pard6. Through use of drugs and elegant photoactivatable proteins, they identify PKC signaling, through RhoA, as a trigger that induces apical actomyosin. Interestingly, apical actomyosin is required for Pard6 accumulation but is not sufficient, indicating the presence of another trigger that allows Pard6 to localize in an actomyosin-dependent manner.

This paper connects several dots to reveal a more coherent pathway for inducing polarity, although several important questions are left unanswered. I found the data high-quality and quantitative, and I agree with most of the interpretations save some as noted below. How polarity is triggered in mammalian embryos is a question of considerable interest to developmental biologists and those interested more generally in cell polarization, so the findings presented should have broad appeal. One point that needs further investigation is the role of myosin (and actin) in Pard6 localization, and I feel that the authors should look at additional markers for Par protein localization. I have several additional minor suggestions for improving the manuscript.

We thank the reviewer for his/her appreciation of the quality of our work and all the positive comments and very useful suggestions that have allowed us to improve our manuscript.

To determine the role of actin in recruiting the Par complex to the apical domain, we have treated embryos with latrunculin B (LatB), a potent actin destabiliser, from early 8-cell stage to the late 8-cell stage. We have now added this data to our revised supplementary Fig. 2c. Briefly, our data show that LatB treatment abolishes apical domain establishment at the late 8-cell stage, supporting that actomyosin organisation is essential for recruiting the Par complex apically (please also see below).

As requested we have also now examined localisation of additional apical polarity markers such as Crb3 and PKC ζ throughout the 8-cell stage and compared their localisation pattern with actomyosin. Similar to our result with Pard6, we found that while F-actin progressively polarised to the apical domain during the early to mid 8-cell stage, Crb3 and PKC ζ remained unpolarised. Only at the late 8-cell stage when F-actin was finally polarised, PKC ζ and Crb3 were localised apically (new supplementary Fig. 1c,g).

Specific point-to-point responses are as follows:

Major comments

1. The negative blebbistatin results are worrisome to me, given that findings are different with inhibitors of myosin activity. How do the authors reconcile these results? One possibility is that inhibitors have varying degrees of effectiveness. Experiments should be performed with all three inhibitors to determine how effectively they block cytokinesis from the 4-8 cell division (I do see evidence of at least some multinucleate cells). Also, experiments should be performed to determine whether F-actin is required for Pard6 apical enrichment.

- 1) We agree with the referee that since actomyosin contractility is very important for cytokinesis, actomyosin inhibitors might be expected to affect cytokinesis in our system. We therefore performed experiments in which three different actomyosin inhibitors were applied from 4-cell stage and embryos were let to develop to the late 8-cell stage. We found that blebbistatin treatment from 4-cell stage led to cytokinesis failure in all of the blastomeres (please see Figure 1 below). In contrast, none of the other inhibitors affected cytokinesis to the equivalent level: ML-7 treatment affected cytokinesis (multi-nucleation) in roughly half of the blastomeres and Y-27632 treatment did not affect cytokinesis. We think that this result confirms the effectiveness of blebbistatin in inhibiting actomyosin contractility, and supports our conclusion that actomyosin contractility is indispensable for apical domain establishment.

Figure 1 for the Referee. Different myosin inhibitors applied from the 4-cell stage, embryos fixed at the late 8-cell stage and examined for the presence of multi-nuclei. Numbers in each bar indicate the number of blastomeres examined and classified into relevant groups.

- 2) We also agree with the referee that it is important to check whether F-actin is required for Par complex polarisation. To address this, we depolymerised F-actin, using a potent actin destabiliser, Lat B. We treated early 8-cell stage embryos (within 1 hr post-cell division) with either DMSO (control) or LatB. Embryos were fixed at the late 8-cell stage and immunostained to reveal Pard6 and determine the establishment of the apical domain. This revealed that LatB treatment disrupted cortical F-actin organisation, and more importantly, Pard6 apical localisation was completely inhibited (please see our revised Fig. 2a-d). We think that this result further supports our conclusion that apical actomyosin organisation is essential for recruiting the Par complex to the apical domain.

2. Basing Par protein behavior on one marker seems problematic, especially considering that there are multiple paralogues of each of the par genes and Par proteins do not always colocalize. aPKC and Par3 should also be examined in key experiments (timing of apical enrichment, dependence on PKC).

We agree with the referee and therefore examined localisation of another Par complex component, PKC ζ , during 8-cell stage and also upon PKC inhibition. We focused on PKC ζ as although our previous results indicated that Par3 also polarises apically at the late 8-cell stage⁵, currently available antibodies do not give a specific signal to examine Par3 localisation confidently. We found that while by the mid-late 8-cell stage (4hrs post-cell division), F-actin already polarised to the cell-contact free surface, PKC ζ remained cytoplasmic and did not show an apical localisation. Only at the late 8-cell stage, PKC ζ became localised apically and was surrounded by the actomyosin ring-like structure (Supplementary Fig. 1c). Upon PLC/PKC inhibition, at the late 8-cell stage, the apically localised PKC ζ signal was lost (Supplementary Fig. 3b,c). These results support our conclusion that 1) actomyosin polarises to the apical domain prior to Par complex apical polarisation. 2) PLC-PKC activity is required for apical Par complex recruitment.

Minor comments

1. Abstract – it was not demonstrated that the Par complex excludes actomyosin to the edge of the apical domain. Language should be softened to suggest that this is the case.

As the referee has advised, we have now revised the abstract accordingly.

2. Introduction – “In contrast to the development of embryos of the majority of metazoan species, mammalian embryos acquire cell polarity *de novo* at a species-specific developmental stage.” I disagree with the first part of this sentence; I can think of many animals that acquire polarity at a discrete point after cleavage has begun.

We thank the reviewer for this suggestion, and have now modified the sentence in the introduction. We state: In contrast to the development of embryos of many species, mammalian embryos acquire cell polarity *de novo* at a species-specific developmental stage.

3. P.3. What do actin and myosin look like at different stages of the 4-cell stage? In other words, could any of their dynamic behavior during the 8-cell stage simply reflect cell-cycle-dependent localization patterns?

We thank the referee for this suggestion. In response, we fixed embryos at early (within 1hrs post-cell division), mid (3-4 hrs post-cell division) and late (8-9 hrs post-cell division) 4-cell stage and immunostained to reveal ppMRLC and F-actin and determine actomyosin activity throughout the 4-cell stage. We quantified apical ppMRLC enrichment. We found no significant difference of ppMRLC nor F-actin cortical localisation (please see our revised supplementary Fig. 2m, n). This result, together with the quantification of 8-cell stage embryos,

indicates that actomyosin is particularly activated at the early 8-cell stage, and the upregulation of actomyosin activity we observe during 8-cell stage is independent of the cell cycle phase.

4. P. 3, first paragraph. Omit “leading to the first cellular asymmetry during embryo compaction.” E-cadherin asymmetry occurs prior to the appearance of actin and myosin asymmetry.

We thank the reviewer for this comment. However, we would like to point out that whether E-cadherin is asymmetric in blastomeres prior to 8-cell stage is rather unclear. Several reports show that E-cadherin only actively polarises at the 8-cell stage. Although immunostaining shows slight enrichment at the cell-contact region before the 8-cell stage, E-cadherin is also distributed at the cell-contact free surface. It is only at the 8-cell stage when apical E-cadherin is actively removed (Fleming et al., 1988)(Clayton et al., 1993). Moreover, the enrichment of E-cadherin at the cell-contact region observed prior to the 8-cell stage is likely due to the overlap of cell membranes, as even in Calcium-free medium E-cadherin levels at the cell-cell contact area remain the same, indicating that such enrichment could be non-functional (Clayton et al., 1993). Overall, we think actomyosin asymmetry serves as the functional symmetry breaking event at the 8-cell stage.

5. What happens to activated myosin upon blebbistatin treatment?

Following the reviewer’s comment, we performed blebbistatin treatment at the early 8-cell stage and examined ppMRLC localisation at the late 8-cell stage. Please see revised Supplementary Fig. 2e,f. We found that upon blebbistatin treatment, although compaction was strongly suppressed, ppMRLC was still enriched at the apical domain. This data indicates that blebbistatin does not inhibit actomyosin activation nor its polarisation.

6. I would not say that Y-27632 treatment “abolished cell polarization” since F-actin still appears polarized (Fig S1e). This needs to be noted in the text.

We agree with the reviewer and have revised our statement. We now state: Y-27632 treatment significantly abolished Pard6’s apical enrichment

7. P. 5-6. It’s unclear why PLC inhibition does not prevent cleavage if it blocks myosin localization. What happens to actin and myosin during cleavage if PLC is inactivated?

We thank the reviewer for his/her comment. We found that upon PLC inhibition some ppMRLC and F-actin still accumulated at the cleavage furrow although it was reduced (please see our attached Figure 2 below). These remaining levels of ppMRLC and F-actin at the cleavage furrow would seem to be sufficient for cytokinesis completion. We think that this suggests that actin can be utilised independently of the PLC pathway in cytokinesis. Indeed although in some organisms, such as *Drosophila*, PLC seems to be required for cytokinesis in some tissues⁶, it does not seem to be the case in other tissues where cytokinesis can take place in the absence of PLC function⁷.

Figure 2 for the referee. 4-cell stage embryos treated with DMSO (control) or U73122 (PLC inhibitor), and cultured until the 4-8 cell stage transition. N=4 for control embryos, N=5 for U73122 treated embryos. Embryos were fixed and immunostained to reveal ppMRLC and F-actin. Yellow arrows indicate the cleavage furrow and accumulated ppMRLC. All scale bars, 15 μ m.

8. Discussion – Several points in the discussion deserve additional attention. How do the authors think that actin and myosin are working to recruit PAR proteins? In *C. elegans*, actomyosin cortical flows carry PAR proteins to the anterior (at the same time), so this mechanism does not fit the authors’ data. Also, what might be the cue that induces PAR asymmetry independently of actomyosin? Finally, how might PKC activity be controlled?

We thank reviewer for pointing out these important points and we agree that it is important to further discuss them.

We now propose that additional scaffolding factors that can bind both actomyosin and the Par complex are involved in recruiting the Par complex apically. Such factors might only appear at the 8-cell stage and therefore premature actomyosin polarisation by itself is insufficient to polarise the Par complex. We have added the relevant discussion in the revised manuscript (the last paragraph).

Our results suggest that cell-contact induced asymmetry of actomyosin is required for Par asymmetry. However, previous findings suggest that microtubules might be able to induce apical domain formation in a small proportion of the cells independent of cell-contact. This alternative route is likely to act as a back-up mechanism⁸. We have added this new concept to the discussion (second paragraph of Discussion).

We agree with the referees that how PKC activity is controlled is an important and open question, which we think right now is very difficult to address, as here we mainly dissect the mechanism of how PKC induces polarisation.

Reviewer #4 (Remarks to the Author):

This is a detailed account of actomyosin symmetry-breaking within the early mammalian embryo, which is an important and underexplored topic with a high level of scientific interest.

There are some beautiful and interesting results shown in the paper but as it stands, there are some major issues that would need resolving before it is suitable for publication. If these issues can be resolved then the paper would be an excellent and important publication for Nature Communications.

I also suggest that the manuscript could be more concise by more grouping of PKC-related experiments for example. This would make it easier to follow and to understand the main conclusions.

We thank reviewer for his/her constructive criticism and his/her appreciation of the quality and importance of this work. We acknowledge the reviewer's comments that the description of PKC-related experiments could be more concise and have therefore revised parts of our manuscript and particularly the description of PKC-related experiments. We hope our revised manuscript addresses all the reviewer's comments.

Specific point-to-point response is as follows:

Major points

1. The authors have not separated compaction and polarisation phenotypes:

E.g. figure 2a ML-7 treated cells: Given the massive difference in cell compaction (and therefore in the general orientation and geometry of the cells) I suggest that this data cannot be used to draw conclusions about the role of actomyosin on cell polarity. To conclude that "MLCK and ROCK-mediated phosphorylation of MRLC is required for actomyosin and Par polarization", the authors would need robust control data demonstrating the alteration of actomyosin and Par polarisation without compaction defects. This issue has not been addressed. E.g. in figure 3h, U73122 causes compaction defects and polarity defects. Both types of defects are rescued by the addition of OAG, making it impossible to distinguish whether the polarity defects were caused by PLC inhibition itself or by compaction inhibition generally.

We thank the reviewer for these comments. The process of compaction and polarisation have been traditionally viewed as mutually interdependent. We agree that to draw a conclusion regarding actomyosin's role in polarisation we need an additional robust control condition in which embryos maintain a compacted geometry. Our rationale for the experimental design is the following: since we observed that actomyosin polarised to the cell contact free surface during compaction (phase I), and Par polarised apically after compaction (phase II), we hypothesized that the inhibition of myosin after completion of phase I should be able to abolish Par complex polarisation without affecting compaction. Therefore, we sought to inhibit myosin activity during the transition between phase I and phase II, to unequivocally determine myosin's role in Par complex polarisation. We chose to inhibit MLCK using ML-7 given that MLCK is the main kinase that phosphorylates MRLC at the 8 cell stage (Fig. 2a-f).

Experimental approach: We first injected GFP-MRLC into 2-cell stage embryos (to visualise myosin localisation and activity) and cultured the embryos until 4-8 cell stage transition. At this point, we monitored each embryo and recorded the time at which embryos entered in the 8-cell stage. When embryos acquired a compacted

morphology (around 4 hours post-cell division) they were transferred to DMSO (control) or ML-7 (MLCK inhibition) medium and cultured until the late 8-cell stage. Embryos were then fixed and analysed.

In both conditions embryos retained a compacted morphology. However, in the ML-7 treated group both GFP-MRLC and F-actin failed to polarise to the cell-contact free surface, and more importantly Pard6 also failed to localise to cell-contact free domain. We have included this data in our revised Supplementary Fig.2i-f. These results support our conclusion that actomyosin apical polarisation is required for Par complex apical recruitment, and that compaction and polarisation are independent processes.

2. The definition of 'apical' and 'basal' is not clear:

This point is related to point 1. Since many of their manipulations affect cell compaction and therefore cell interaction and arrangement, I worry that the authors have not addressed how this might affect the axis of polarity within these cells. E.g. supplementary figure 6e: RhoA is still activated in a polarised manner but in a different subcellular location. I don't think it is appropriate to define the contact-free edges of the cells as 'apical' in situations where compaction is prevented. There are also examples where there appear to be centrally-located accumulations of 'apical' proteins (e.g. the late 8-cell-2 in figure 2g has a centrally located ring of ppMRLC and there is a centrally located accumulation of F-actin and Pard6 in Y-27632 treated cells in supplementary figure 1e). What counts as 'apical' in these cases?

The distinction between 'apical' and 'basal' is especially important given the author's analyses of intensity levels from these regions. If there is apical domain localisation at the centre of the embryo but this is being classed as 'basal' then this would confuse the results. The authors should clearly illustrate how they sample their data (e.g. show the ROIs used to measure mean intensity levels).

We agree with the reviewer that in the absence of compaction and altered polarisation it is difficult to define the contact-free domain as apical. Therefore, we have used "cell-contact free surface" to denote the membrane domain in contact with the outside. We also agree with the reviewer that although we found in many situations that the polarity components failed to localise to the cell contact-free domain, we cannot necessarily draw the conclusion that polarity is abolished, but rather it is possible that the polarisation axis is reversed. Bearing this in mind, we have re-examined all our data, and re-analysed the signal intensity of Pard6/F-actin to determine the ratio of cell-contact versus cell-contact free domains, using a circumferential analysis method as suggested by the reviewer (attached Figure 3). Essentially, in all of the inhibitor treatments the level of enrichment in either the cell-contact or cell-contact free domain is significantly lower than the apical enrichment in natural 8-cell stage polarisation. In addition, we found no consistent tendency of signal enhancement towards any of the domains. Therefore, these results indicate that PLC-PKC inhibition as well as RhoA inhibition abolish apical-basal polarity. Regarding the observation that F-actin and ppMRLC sometimes localise centrally in late 8-cell stage embryos, we have carefully examined this, and observed that it can be seen only in a proportion of 8-cell stage embryos. Since it has been previously reported that at the 8-cell stage actin and myosin are involved in the formation of other cell structures such as filopodia, we propose that this transient enrichment could come from such compartments.

Finally, we agree with the reviewer that it is important to clearly state how we sample the data. Therefore, we have included our image processing method in our revised supplementary Fig.1b. Briefly, we used the freehand line tool in Fiji (line width = $0.8\mu\text{m}$, so the line can cover all the cortical region); we drew along the cell-contact free surface or cell-cell contact region and defined the ROI. ROIs were selected from the same imaging plane/z-step to avoid the "objective-distance" effect. Please refer to our revised Fig. 1b and Supplementary Fig.1e for the detailed method.

Figure 3 for the referee. a) Pard6 signal intensity ratio between cell-contact free and cell-contact region in blastomeres treated with DMSO or PLC-PKC inhibitors. b) Pard6 signal intensity ratio between cell-contact and cell-contact free region in blastomeres treated with DMSO or PLC-PKC inhibitors. The blue line indicates the position of “value =1”.

3. Line analyses are not always appropriate

Polarity proteins such as Pard6 often initially appear in ‘spot’-like clusters, rather than being “distributed equally around the cell cortex”. This appears to be the case here (see spots of Pard6 around the cell cortex in figure 1a). This makes a line analysis inappropriate and brings into question the result that actomyosin polarisation preceded Pard6 polarisation. Could the authors also analyse intensity around the circumference of cells in figure 1a. This would also aid in the analysis of ‘actin ring formation’, which the authors mention but which is not possible to see clearly in figure 1a (it’s a bit clearer in the example shown in figure 5a but an analysis would still help).

Also, what determines which cell polarises first? Are the authors consistent with which cell they select for analysis? E.g. in figure 1d, Pard6 is clearly polarised by the mid-late 8-cell stage and on the second cell appears polarised in relation to background levels even at the mid 8-cell stage but this cell has not been analysed.

We greatly thank the reviewer for his/her input regarding the data analysis. We have followed the reviewer’s suggestion and have re-displayed our quantification. Please see new Fig.1b and Supplementary Fig. 1e. Briefly, we drew a line (width = $0.8\mu\text{m}$) to cover the whole cell-contact free domain or cell-contact domain, and plotted the signal of corresponding domain (as illustrated in Supplementary Fig. 1b). We have now applied the quantification method suggested by the reviewer to quantify the apical enrichment of actomyosin and Par polarity proteins. We have revised our material and methods section to clarify this point.

The reviewer correctly points out that one of the features of 8-cell stage polarisation is the asynchrony. What determines the cell that polarises first is an interesting and an important question to address but we think that addressing this question is beyond the scope of our manuscript. Regarding the heterogeneity in the timing of polarisation, we have taken into account this point by analysing all the blastomeres in mid-late to late 8-cell stage embryos. This is displayed in revised Fig. 1c and Supplementary Fig. 1f.

4. The sufficiency of actomyosin asymmetry to polarize Par complexes is not demonstrated:

To say that “asymmetry of the functional actomyosin network is required for Par complex polarity” (i.e. it is necessary) would require a robust compaction control (see point 1 above). Even so, to say that “induction of this cytoskeletal asymmetry is an absolute pre-requisite for the cortical enrichment of the Par polarity complex to establish cell polarity” would require the sufficiency of actomyosin asymmetry to mediate Par asymmetry. This has not been shown. However, some of the data that the authors say does not show an upregulation of Pard6 seems like it actually does. E.g. it looks to me like there is a small upregulation of Pard6 in the OAG treated embryo shown in figure 5c. This is quantified in supplementary figure 4C and is significantly different. Why do the authors say that there is no apical enrichment of Pard6? Similarly, in figure 5f, there is a polarised enrichment of Pard6 in the cells injected with PKCa-A25E. However, this enrichment is in opposite sides of the 2 injected cells and these cells have different levels of construct expression making it hard to interpret this data.

We thank reviewer for the comment, and agree that there seems to be a small enrichment of Pard6 in the exemplified picture. We have carefully examined all of the embryos in the PKCa-A25E overexpression experiment and found that there is no consistent pattern of Pard6 enrichment to the cell-contact free or cell-contact surface (please see our revised supplementary Fig. 5c). In some embryos Pard6 seems slightly enhanced to the apical region, whereas in other embryos the signal seems stronger in the basal. In both cases, the signal intensity is fairly weak compared to the natural polarisation happening at the 8-cell stage. The same applies to OAG treated 4-cell stage embryos. Careful quantification of Pard6 fluorescence intensity showed no significance difference in control embryos compared to OAG treated embryos (please see our revised supplementary Fig. 5d). Collectively, we think it is reasonable to state that Pard6 remains unpolarised upon ectopic PKC activation.

Regarding the sufficiency of actomyosin in recruiting Par complex, we totally agree with reviewer that it is important to address this point to fully reveal the role of actomyosin in polarising Par complex. To this end, we used three different methods to trigger actomyosin’s apical polarity: OAG treatment at the 4-cell stage; over-expression of PKCa-A25E; and induction of PKC through optogenetics. In none of these experiments does Pard6 show premature apical localisation, despite activation of actomyosin (please also see preceding paragraph). On the contrary, when actomyosin is ectopically activated at late 8-cell stage onwards, the Pard6 region is expanded to the region where actomyosin is ectopically activated (OAG 8-cell stage treatment, Fig. 5a,b). We propose that actomyosin itself is insufficient to directly trigger Par complex binding and that additional proteins are likely to be required, and these additional proteins could only be present at the 8-cell stage. However, as multiple ways of

inhibiting actomyosin consistently down-regulated Pard6's apical association, we believe it is reasonable to conclude that actomyosin apical activation is an important pre-requisite for the Par complex's apical recruitment that happens later.

5. Optogenetic approaches need further analysis:

The author's optogenetic experiments allow them to directly determine the causative links in the molecular cascades. However, this data requires further explanation and analysis:

We thank the reviewer for pointing out all the concerns, and have provided additional analysis following his/her comments.

- figure 5i-k and video 5: why is CIB1-CAAX only localised to one part of the cell? Is it the specific ROI that denotes the subcellular specificity of PKC activation or is it due to the subcellular localisation of CIB1-CAAX and how was this achieved?

We apologise for not describing this point clearly in the original manuscript. In our optogenetic experiment we defined a specific ROI (by drawing a fixed oval area) in the cell-contact free surface of a 4-cell stage blastomere. Consecutive laser pulses at 458nm were applied on the specific area, and the detector was set to detect GFP (CIB1-CAAX) localisation. Following each pulse, another scan at 568nm was used to detect RFP (PKC-kinase domain) signal. As the 458nm laser pulse was only applied in the ROI region, the CIB1-CAAX signal is only observed in the ROI (despite the fact that CIB1-CAAX localises all around the cell membrane). By doing this we achieved specific PKC activation in a defined subcellular area. We have added a more detailed description of the optogenetic experiment in the "materials and methods" section to avoid confusion.

- the authors should explain why the recruitment of the optogenetic effectors (kinase domain of PKC and RhoA L63-C190R) to the membrane causes specific activation and how they have avoided background levels of activation.

We apologize for the lack of clarity in the original version. In the original design of the constructs we took into consideration the background activation and devised a strategy to minimize it.

PKC construct: The rate-limiting step for PKC activation is its translocation to the membrane. Therefore, we fused the PKC kinase domain to the CRY2-mCherry optogenetic construct (CRY2-mCherry-PKC-KD) in the absence of a membrane localisation signal. In the absence of blue light or CIB1-CAAX the construct remains in the cytoplasm (Please refers to our supplementary Fig. 6c,d). To better demonstrate the specificity of the construct and to show that PKC only gets activated in the ROI, we have now quantified ppMRLC levels outside the ROI or in cells that are not exposed to the blue light. Moreover, we also constructed a CRY2-mCherry construct without a PKC kinase domain. Using the same illumination setting as for CRY2-mCherry-PKC-KD, we observed that CRY2-mCherry also translocated to the membrane, but was incapable of upregulating ppMRLC. Together, these control conditions suggest that the design of our construct minimises background activation of PKC and allows a localised and specific activation of PKC (Please refers to our revised supplementary Fig. 6a-d).

RhoA construct: We introduced RhoA Q63L mutation to keep RhoA constitutively active when it localised to the membrane. In addition, and more importantly, we introduced another mutation - the C190R. This mutation abolished RhoA membrane localisation and therefore RhoA-Q63L-C190R cannot activate actomyosin, unless there is an additional condition provided that induces membrane localisation of RhoA^{9,10}. To show that using our design, we can indeed control RhoA activation in our region of interest, we compared the level of ppMRLC in the region where CRY2-mCherry-RhoA-Q63L-C190R translocated to the membrane and the region where there is no membrane localisation of CRY2-mCherry-RhoA-Q63L-C190R, and also compared this to CRY2-mCherry only construct. We found that the region that has no RhoA-membrane translocation shows a similar level of ppMRLC as the CRY2-mCherry translocated region, and significantly lower than a RhoA activated region. Collectively, these results suggest that our RhoA optogenetics construct can trigger specific RhoA activation in the region of interest. (please refers to our revised supplementary Fig. 6d).

Also, how did they achieve reproducible levels/control for different levels of each of the heterodimerization components between experiments? E.g. Sup fig 6g: The levels of CRY2-RhoA-L63-C190R are very different between the 2 panels. How do the authors control for this and be sure that RhoA is not activated without cortical localisation? There is no description either in the figure legend or main text/methods for how RhoA is specifically activated using this method.

As the referee requested, we have now added the relevant description in materials and methods. To achieve reproducible levels of optogenetic constructs we injected the same concentration of the mRNAs. Since the microinjection amount can be slightly different for each blastomere, the expression level of each cell can marginally differ causing some variability of CRY2 translocation and activation effect. To address this we now provide the quantification of activated PKC region and compared with two different control conditions: a) non-

activated PKC region and b) CRY2-mCherry only (without PKC kinase domain) with the quantification of pooled activated blastomeres, in which constructs are injected at the same concentration as PKC/RhoA (Supplementary Fig. 6c,d). Our results indicate that the consistent upregulation of ppMRLC/F-actin is specific to the local activation of PKC/RhoA and it is absent in control conditions.

- Results should be quantified and the authors should show all 3 channels (e.g. in figure 5: CIB1-CAAX, CRY2-PKC-KD and F-actin/ppMRLC/Pard6) levels before and after optical stimulation to control for background activation and for intrinsic differences in individual cell polarisation. Illumination ROIs should be clearly defined and made distinct from boxes denoting magnified regions (e.g. figure 5J and K).

We thank the reviewer for his/her suggestion and have added additional information/images to the figure (Fig. 5j, k and Supplementary Fig. 5a-d).

- Internal control cells should also be quantified

We have quantified the internal control cells for all the control conditions (revised supplementary Fig. 5d).

- I'm not sure that the Cryptochrome reference used (Liu et al 2008) is the most appropriate in isolation. It would be a good idea to also include a paper that shows the use of these proteins for experimental heterodimerization. I believe the first example of this for the CRY2 system is Kennedy et al 2010 Nature Methods).

We thank the reviewer for this suggestions and now cite the proposed paper.

6. The authors do not show any data demonstrating that apical actomyosin initiates 'fate decisions'. This should therefore be removed from the title.

We agree with the reviewer and have changed our title.

Minor points

- It is important for the authors to be specific about what they mean by 'polarity' throughout the paper. Eg. First line of page 9 "this apical localisation is necessary for the correct establishment of cell polarity at the 8-cell stage". The data in figure 6 does not address Par6 polarity so this is only referring to actomyosin polarity.

We thank the reviewer for pointing this out and have therefore revised this sentence.

- The authors should define acronyms on first use: e.g. PLC

We have corrected this in the manuscript.

- Methods: For microinjection the total amount of RNA injected should be listed, not just the concentration of solution.

We thank the reviewer for this comment and have added the information.

- In the 'statistics' section, first line: 'qualitative data' is the wrong terminology here since qualitative data cannot be statistically analysed.

Qualitative data can be statistically analysed using the appropriate statistical tests (Chi2 and Fisher's exact test). We would like to refer the reviewer to the following publication: Bewick et al, 2004, Crit Care¹¹.

Specific figure points

We thank the reviewer for all the following comments. We have corrected all the points below accordingly.

Figure 1

- In Figure 1a, panels A and B need labelling. Perhaps use i and ii to distinguish from figure 1a and b?

- To see the actin ring in figure 1a more clearly, the actin 3D channel should be shown alone

- I think there may be an issue with directly comparing the fluorescence levels of Pard6 and F-actin since levels of F-actin are higher from the start. Perhaps a better read-out would be the normalised fluorescence.

- Can the analyses in Figure 1c and 1f (as well as similar analyses throughout the paper) be statistically analysed?

- The white read out in the MRLC channel makes it hard to interpret the projection. This is the case throughout the paper.

- Figure 1g: colours are too similar – hard to distinguish GFP-MRLC from background of cell body. I like the upper panel but the lower panel doesn't accurately reflect the basolateral localisation of F-actin and GFP-MRLC so could be a little confusing. Perhaps it is not required?

Figure 2

- See major points 1 and 2 above
- Figure 2a: I cant see the Pard6 staining in the first and last panels, even though this was visible at similar stages in figure 1.
- Could the lack of effect of blebbistatin on polarisation be due to the later treatment?
- Figure 2f – should explain in legend how apical enrichment values are calculated. Same for figure 4b,d,f and h

Figure 3

- Supplementary videos 3 and 4 should be labelled.

Figure 5

- See major points 4 and 5 above
- Figure 5b – explain the y-axis somewhere obvious, like the figure legend.
- Figures 5c and 5f: The authors should clarify what embryonic stage these experiments occur at – 4 cell stage?
- The reference to supplementary figure 3d should be supplementary fig 4d.

Figure 6

- Supplementary fig 6e – see major point 2 above.
- I found this section quite hard to follow. A diagram of hypotheses and conclusions would help.

References for response:

1. Fierro-Gonzalez, J.C., White, M.D., Silva, J.C. & Plachta, N. Cadherin-dependent filopodia control preimplantation embryo compaction. *Nature cell biology* 15, 1424-1433 (2013).
2. Maitre, J.L., Niwayama, R., Turlier, H., Nedelec, F. & Hiiragi, T. Pulsatile cell-autonomous contractility drives compaction in the mouse embryo. *Nature cell biology* 17, 849-855 (2015).
3. Maitre, J.L. et al. Asymmetric division of contractile domains couples cell positioning and fate specification. *Nature* 536, 344-348 (2016).
4. Bloom, T. & McConnell, J. Changes in protein phosphorylation associated with compaction of the mouse preimplantation embryo. *Molecular reproduction and development* 26, 199-210 (1990).
5. Plusa, B. et al. Downregulation of Par3 and aPKC function directs cells towards the ICM in the preimplantation mouse embryo. *Journal of cell science* 118, 505-515 (2005).
6. Wong, R., Fabian, L., Forer, A. & Brill, J.A. Phospholipase C and myosin light chain kinase inhibition define a common step in actin regulation during cytokinesis. *BMC cell biology* 8, 15 (2007).
7. Wong, R. et al. PIP2 hydrolysis and calcium release are required for cytokinesis in *Drosophila* spermatocytes. *Current biology* : CB 15, 1401-1406 (2005).
8. Houliston, E., Pickering, S.J. & Maro, B. Alternative routes for the establishment of surface polarity during compaction of the mouse embryo. *Developmental biology* 134, 342-350 (1989).
9. Miyazaki, K., Komatsu, S. & Ikebe, M. Dynamics of RhoA and ROKalpha translocation in single living cells. *Cell biochemistry and biophysics* 45, 243-254 (2006).
10. Kranenburg, O., Poland, M., Gebbink, M., Oomen, L. & Moolenaar, W.H. Dissociation of LPA-induced cytoskeletal contraction from stress fiber formation by differential localization of RhoA. *Journal of cell science* 110 (Pt 19), 2417-2427 (1997).
11. Bewick, V., Cheek, L. & Ball, J. Statistics review 8: Qualitative data - tests of association. *Critical care* 8, 46-53 (2004).

Reviewers' Comments:

Reviewer #1:

Remarks to the Author:

The authors have addressed all my questions and comments satisfactorily.
I therefore fully support publication of this highly interesting study.

Reviewer #2:

Remarks to the Author:

I am satisfied with the revised manuscript.

Reviewer #3:

Remarks to the Author:

The authors have addressed all of my concerns with the original manuscript, and I support publication in Nature Communications.

Reviewer #4:

Remarks to the Author:

The authors have made their manuscript clearer and I believe that the analyses are now more robust and that the manuscript is suitable for publication in Nature Communications. I congratulate the authors on a beautiful piece of work. Specific comments are below:

1. The data in supplementary figure 2i-l is much more compelling and provides a suitable control showing alteration of actomyosin and Par polarisation without compaction defects. I am still concerned by the large compaction phenotypes seen elsewhere in the paper – e.g. supplementary figure 7. These could be more clearly addressed.
2. The re-definition of domains and circumferential analysis is clearer
3. The analysis is clearer now
4. The extra analysis in supplementary figure 5 clarify the conclusion that ectopic PKC activation does not cause polarisation of Par6. The author's reasoning for the necessity of actomyosin for Par apical recruitment makes sense although the difference between necessity and sufficiency could be made clearer in the introduction. These results do leave open the question of what it is that recruits Par6 to the membrane if actomyosin is not sufficient but this point has now been clearly addressed in the discussion.
5. Supplementary figure 6 makes the optogenetic strategy much clearer. Can you label supplementary figure 6a and b with 'F-actin' and 'ppMRLC'? also, I wonder if one of the lower panels in supplementary figure 6a (second from right) is the correct picture? It seems very different to its neighbouring panels and more like the one above it. The labelling on the other optogenetic figures is clearer now.

REVIEWERS' COMMENTS:

Reviewer #1 (Remarks to the Author):

The authors have addressed all my questions and comments satisfactorily. I therefore fully support publication of this highly interesting study.
We thank the reviewer for supporting our work for publication in Nature communications.

Reviewer #2 (Remarks to the Author):

I am satisfied with the revised manuscript.
We thank again the reviewer for all the helpful comments and his support for the publication of our manuscript.

Reviewer #3 (Remarks to the Author):

The authors have addressed all of my concerns with the original manuscript, and I support publication in Nature Communications.
We thank the reviewer for all the helpful suggestions and his support for the publication of our manuscript.

Reviewer #4 (Remarks to the Author):

The authors have made their manuscript clearer and I believe that the analyses are now more robust and that the manuscript is suitable for publication in Nature Communications. I congratulate the authors on a beautiful piece of work. Specific comments are below:

We thank the reviewer for his helpful comments. Please see our point-by-point response to the remaining concerns:

1. The data in supplementary figure 2i-l is much more compelling and provides a suitable control showing alteration of actomyosin and Par polarisation without compaction defects. I am still concerned by the large compaction phenotypes seen elsewhere in the paper – e.g. supplementary figure 7. These could be more clearly addressed.

Following the reviewer's suggestion, we have now discussed this point in our revised discussion – we invite the reviewer to see our revised manuscript page 12 line 3.

2. The re-definition of domains and circumferential analysis is clearer

3. The analysis is clearer now

4. The extra analysis in supplementary figure 5 clarify the conclusion that ectopic PKC activation does not cause polarisation of Pard6. The author's reasoning for the necessity of actomyosin for Par apical recruitment makes sense although the difference between necessity and sufficiency could be made clearer in the introduction. These results do leave open the question of what it is that recruits Par6 to the membrane if actomyosin is not sufficient but this point has now been clearly addressed in the discussion.

5. Supplementary figure 6 makes the optogenetic strategy much clearer. Can you label supplementary figure 6a and b with 'F-actin' and 'ppMRLC'? also, I wonder if one of the lower panels in supplementary figure 6a (second from right) is the correct picture? It seems very different to its neighbouring panels and more like the one above it. The labelling on the other optogenetic figures is clearer now.

The signal for the movie snapshots displayed in Supplementary Figure 6a and b are from the optogenetic constructs – CRY2-PKC-KD and CIBN-CAAX. Panels 6a and 6b do not show F-actin and ppMRLC.

We have corrected the redundant image in supplementary 6b as reviewer pointed out.